# Immunosenescence and vaccine efficacy revealed by immunometabolic analysis of SARS-CoV-2-specific cells in multiple sclerosis patients

Sara De Biasi [1,8] ✉, Domenico Lo Tartaro[1,8], Anita Neroni[1], Moritz Rau [1,2], Nikolaos Paschalidis [3], Rebecca Borella[1], Elena Santacroce[1], Annamaria Paolini[1], Lara Gibellini [1], Alin Liviu Ciobanu [1], Michela Cuccorese[4], Tommaso Trenti[4], Ignacio Rubio [2], Francesca Vitetta[5], Martina Cardi [5], Rafael José Argüello [6], Diana Ferraro[5] & Andrea Cossarizza [1,7] ✉

Disease-modifying therapies (DMT) administered to patients with multiple sclerosis (MS) can influence immune responses to SARS-CoV-2 and vaccine efficacy. However, data on the detailed phenotypic, functional and metabolic characteristics of antigen (Ag)-specific cells following the third dose of mRNA vaccine remain scarce. Here, using flow cytometry and 45-parameter mass cytometry, we broadly investigate the phenotype, function and the single-cell metabolic profile of SARS-CoV-2-specific T and B cells up to 8 months after the third dose of mRNA vaccine in a cohort of 94 patients with MS treated with different DMT, including cladribine, dimethyl fumarate, fingolimod, interferon, natalizumab, teriflunomide, rituximab or ocrelizumab. Almost all patients display functional immune response to SARS-CoV-2. Different metabolic profiles characterize antigen-specific-T and -B cell response in fingolimod- and natalizumab-treated patients, whose immune response differs from all the other MS treatments.

The immunosuppressive and immunomodulatory disease-modifying therapies (DMT) used for multiple sclerosis (MS) act at different levels, i.e., inhibiting the expansion of activated lymphocytes (teriflunomide), redirecting pathological immune cells away from the central nervous system [natalizumab, fingolimod (FTY)] or depleting immune cell subsets (B and T cells; anti-CD20, cladribine)[1]. In treated patients, DMT can introduce risk for increased infections, reduced vaccine effectiveness or reduce the duration of specific immunity. These aspects are of critical importance, especially in the course of a pandemic such as that due to SARS-CoV-2, where the host immune response is crucial[2–12], and that was effectively fought by several different vaccines.

In patients with MS, DMT such as interferon (IFN)-β, glatiramer acetate and dimethyl fumarate (DMF) are not expected to compromise

[1]Department of Medical and Surgical Sciences for Children and Adults, University of Modena and Reggio Emilia School of Medicine, Modena, Italy. [2]Department of Anesthesiology and Intensive Care Medicine, Jena University Hospital, Jena, Germany. [3]Biomedical Research Foundation Academy of Athens, Athens, Greece. [4]Department of Laboratory Medicine and Pathology, Diagnostic Hematology and Clinical Genomics, Azienda Unità Sanitaria Locale AUSL/AOU Policlinico, Modena, Italy. [5]Neurology Unit, Department of Biomedical, Metabolic and Neurosciences, Nuovo Ospedale Civile Sant'Agostino Estense, University of Modena and Reggio Emilia, Modena, Italy. [6]Aix Marseille Univ, CNRS, INSERM, CIML, Centre d'Immunologie de Marseille-Luminy, Marseille, France. [7]National Institute for Cardiovascular Research, Bologna, Italy. [8]These authors contributed equally: Sara De Biasi, Domenico Lo Tartaro. ✉e-mail: sara.debiasi@unimore.it; andrea.cossarizza@unimore.it

vaccine efficacy[13], although the effect of DMF-induced lymphopenia on vaccine efficacy is unknown, and attenuated vaccine responses in patients with moderate or severe lymphopenia is conceivable[14]. A modestly diminished rate of immune response to vaccines was described in patients treated with teriflunomide, even if this did not compromise the achievement of seroprotective antibody levels[15]. Valid immune response to diphtheria-tetanus toxoid and to Keyhole limpet hemocyanin (KLH) was found in natalizumab-treated patients[16], while H1N1 and seasonal influenza vaccination provided evidence that an adequate response to the immunization may not occur in some patients[17,18]. Adequate immune responses to seasonal influenza vaccine and tetanus toxoid booster were detected in patients receiving FTY[19]. On the other hand, MS patients treated with cell-depleting agents (such as ocrelizumab, rituximab, ofatumumab, alemtuzumab, and cladribine) displayed attenuated vaccine responses, especially if they were vaccinated during the maximum cell depletion period. Peripherally B cell-depleted ocrelizumab recipients mounted attenuated humoral responses to clinically relevant vaccines and the neoantigen KLH, suggesting that use of standard non-live vaccines while on ocrelizumab treatment requires careful considerations[20]. It is nevertheless recommended to vaccinate patients for seasonal influenza because a potentially protective humoral response, even if attenuated, can be expected[21].

How different DTM affect vaccination effectiveness and safety in patients with MS was highlighted during the outbreak of coronavirus diseases (COVID-19). In particular, therapies with anti-CD20 (rituximab/ocrelizumab) monoclonal antibodies or with the sphingosine-phosphate receptor modulator (FTY) have been shown to weaken the formation of immune response after SARS-CoV-2 vaccination[22–31]. MS patients treated with teriflunomide or alemtuzumab achieved effective humoral and cellular immune responses up to 6 months following the second COVID-19 vaccination. Immune responses were reinforced following the third vaccine booster[32]. However, the response to vaccination was mainly measured by humoral responses (in term of antibody titers in plasma) and or production of interferon (IFN)-γ by T cells as correlate for a protective response. However, the protective capacity of the adaptive immune response to SARS-CoV-2 depends not only on virus-specific antibodies, but also on the cellular response[33]. The phenotype of antigen-specific (Ag+) T cells of patients treated with rituximab/ocrelizumab displayed a skewed response, mostly compromising circulating follicular helper T (Tfh) cell responses and augmenting the induction of CD8+ T cell[33]. Moreover, when compared to healthy donors (HD), MS patients showed lower percentages in Ag-specific cells able to produce IFN-γ, interleukin (IL)−2 and tumor necrosis factor (TNF)[34].

A detailed overview of different functional and metabolic features of the long-term immune response after vaccination in relapsing-remitting (RR) MS patients treated with different DMT is still missing. Here, we broadly interrogate SARS-CoV-2 antigen-specific T and B cells 6 months after the third dose of mRNA vaccine in a cohort of 94 MS patients treated with different DMT such as cladribine, DMF, FTY, IFN-β, natalizumab, teriflunomide or rituximab/ocrelizumab. By using 21-parameter flow cytometry, we investigate the phenotype and function of antigen-specific T and B cells. In addition, the metabolic profile of such cells is examined using 45-parameter mass cytometry, allowing profiling of the metabolic regulome at the single-cell level (scMEP)[9,35,36]. We find that almost all patients develop a detectable and functional SARS-CoV-2 immune response. In particular, we show that a diverse metabolic profile characterizes antigen-specific T and B cell response in FTY- and natalizumab-treated MS patients, who generate a unique immune response that differs from all other MS treatments. Finally, using our own approach of prediction analysis, we identify a SARS-CoV-2 specific immunological signature that could likely predict protection from breakthrough SARS-CoV-2 infection.

## Results

### Demographic and clinical characteristics of the patients

MS patients and healthy donors had a median age of 44.0 (interquartile range, IQR: 41.5–48.5), were mostly female (71.7%), with a median disease duration of 14.3 years (IQR: 10.0–17.1). The most common anti-COVID-19 vaccine used was Pfizer-BioNTech (Comirnaty): 69 persons (64.2%), followed by Moderna (Spikevax): 38 persons (35.8%). Median time from the last dose of vaccine to sample collection was 4.4 months (IQR: 3.8–5.3). Demographic and clinical characteristics of 94 MS patients and 13 healthy donors (HD), the type of DMT at the time of vaccination, the type of third dose vaccine and median range of time to last administration, prior COVID-19 infection status, and relevant comorbidities are shown in Supplementary Data 1. Patients were eligible for inclusion if they met the following criteria: (a) a confirmed diagnosis of Relapsing-Remitting Multiple Sclerosis (RRMS), and (b) a history of treatment with FTY, dimethyl fumarate, natalizumab, or teriflunomide for a minimum of 6 months, or having undergone at least two infusion cycles with rituximab or ocrelizumab or completed at least one full cycle of cladribine. Patients on ocrelizumab or rituximab, as per routine clinical practice, underwent SARS-CoV-2 vaccination at least six weeks before subsequent infusion or at least three months after the last infusion. Exclusion criteria comprised treatment with steroids during the preceding 6 weeks and a history of COVID-19 before vaccination. Patients treated with different DMT were enrolled such as: natalizumab (n = 15; 15.9%), DMF (n = 18; 19.1%), DMF patients with decreased absolute lymphocyte counts (<800/μL) at the time of sampling, defined "DMF lymphopenic" (n = 10; 10.6%), interferon IFN (n = 12; 12.8%), FTY (n = 14; 14.9%), rituximab/ocrelizumab [n = 11; 11.7%, which included those treated with ocrelizumab (n = 7; 63.6%) or rituximab (n = 4; 36.4%)], cladribine (n = 6; 6.4%), and teriflunomide (n = 8; 8.6%).

### MS patients treated with different DMT develop similar percentages of Ag+ CD4+ T cells, but these cells display a different phenotype compared to healthy donors

First, we investigated by manual gating the percentage of CD4+ T cells. Figure 1A shows that lymphopenic patients treated with DMF displayed higher percentage of CD4+ T cells if compared to HD, while all other MS patients treated with different therapies showed similar percentages of CD4+ T cells. The absolute number of CD4+ T cells was lower in patients treated with cladribine, FTY- and lymphopenic-DMF treated patients and higher in those treated with natalizumab if compared to HD.

Then, we identified Ag+ T cells, defined as cells expressing CD137 and CD69 after 18 h of in vitro stimulation with SARS-CoV-2 peptides (Supplementary Fig. 1)[9,35]. As shown in Fig. 1B, the percentage of Ag+ T cells within CD4+ T cells was similar in all MS patients and HD, confirming that MS patients treated with different drugs mount a detectable specific T cell response. The absolute number of Ag+ CD4+ T cells was lower in FTY-treated patients if compared to HD and DMF-, IFN-, natalizumab-, teriflunomide- or rituximab/ocrelizumab-treated patients.

Then, the pool of Ag+ CD4+ T cells was analyzed by an unsupervised method, i.e., FlowSOM, to better depict their phenotype in terms of differentiation and T helper polarization towards circulating follicular helper (Tfh), Th0/Th2, Th1 or Th17 (Fig. 1C, Supplementary Fig. 2). Cell clustering resulted in 15 different populations that represent the entire differentiation spectrum, from the most undifferentiated cell type (i.e., naïve) to the most differentiated one, such as effector memory T cells (EM). The expression of antigens such as CD45RA, CCR7, CD27, CD28 and CD95 was used to identify the differentiation status to define naïve (CD45RA+CCR7+CD27+CD28+CD95−), stem memory cell (T_SCM, CD45RA+CCR7+CD27+CD28+CD95+), central memory (CM, CD45RA−CCR7+CD27+CD28+CD95+), transitional memory (TM, CD45RA−CCR7−CD27+CD28+CD95+), and effector memory (EM) (CD45RA−CCR7+CD27+CD28−CD95+). Surface molecules such as

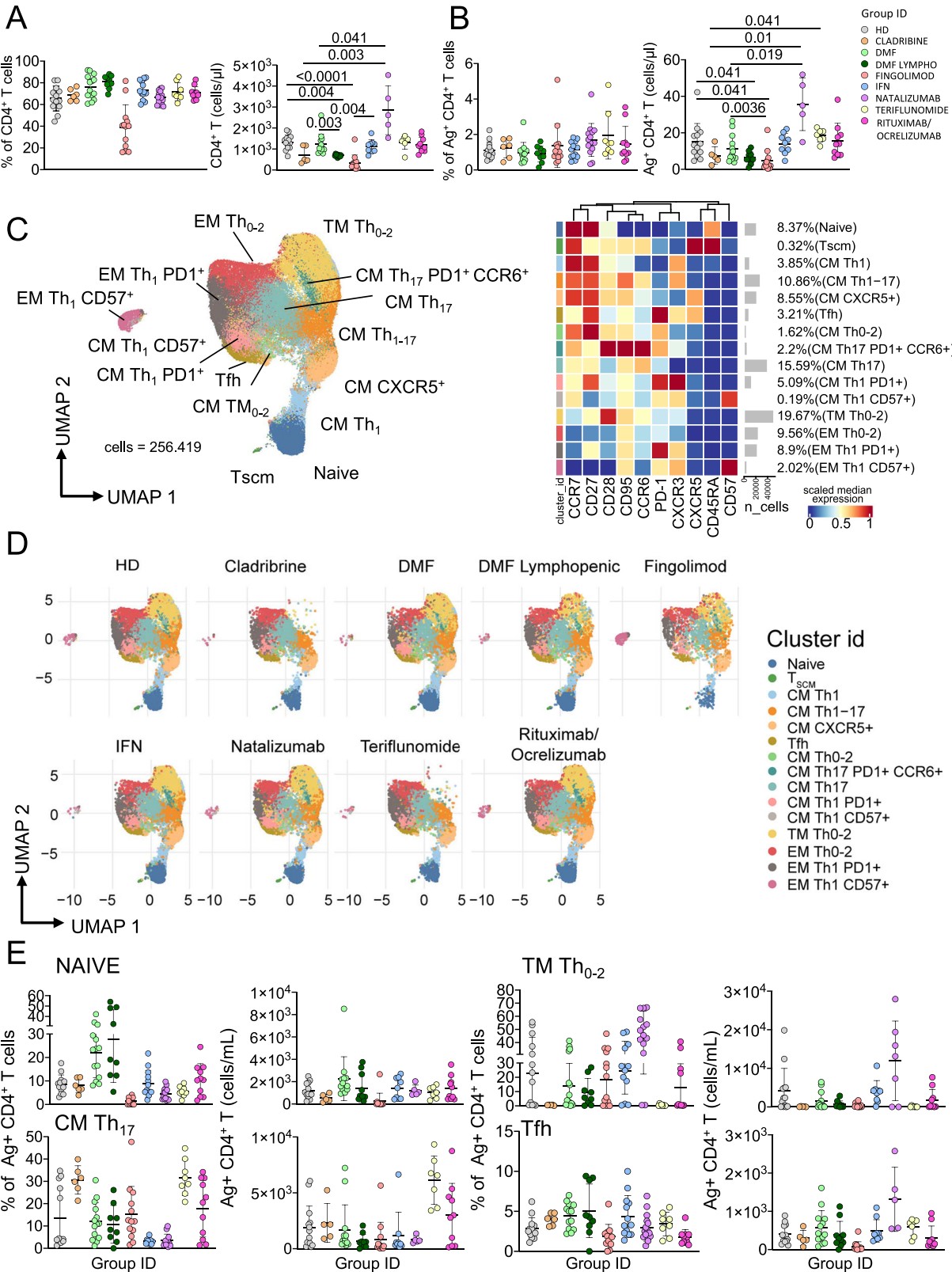

CXCR3, CCR6, CXCR5 and PD1 were used for a classification regarding Th polarization, i.e., Th0/Th2 defined as CXCR3−CCR6−, Th1 as CXCR3+CCR6−, Th17 as CXCR3−CCR6+, Th1/Th17 as CXCR3+CCR6+ and Tfh as CXCR3−CCR6−CXCR5+PD1+. Senescent T cells were characterized by the expression of CD57 (Fig. 1C).

Even if MS patients mount a CD4+ Ag+ T cell response whose frequency was similar among HD and MS patients treated with different drugs, cellular composition was phenotypically different. Differences were evident for the percentage of naïve cells and the Th1 compartment as far as DMF- or teriflunomide-treated patients are concerned (Fig. 1D), but these differences were lost when absolute numbers were considered (Fig. 1E). Natalizumab-treated patients displayed the highest percentage of Ag+ TM Th0-2 cells if compared to healthy donors or to the other treatments; this difference was maintained also

**Fig. 1 | Ag⁺ CD4⁺ T cell landscape. A** Percentage and absolute number of CD4⁺ T cells. Dot plots show the percentage and absolute number of CD4⁺ T cells. The central bar represents the mean ± SD. Kruskal–Wallis test (one-sided) with Benjamini–Hochberg correction for multiple comparisons. Significant adjusted q-values are reported in the figure. **B** Percentage and absolute number of Ag⁺ CD4⁺ T cells. Dot plots show the percentage and absolute number of Ag⁺ CD4⁺ T cells. The central bar represents the mean ± SD. Kruskal–Wallis test (one-sided) with Benjamini–Hochberg correction for multiple comparisons. Significant adjusted q-values are reported in the figure. **C** Ag⁺ CD4⁺ T cells phenotype UMAP and Heatmap. Uniform Manifold Approximation and Projection (UMAP) plot shows the 2D spatial distribution of 256.419 cells from healthy donors (HD) and MS patients treated with different DMT embedded with FlowSOM clusters. Heatmap of the median marker intensities of the 10 lineage markers across the 15 cell populations obtained with FlowSOM algorithm after the manual metacluster merging. The colors of cluster_id column correspond to the colors used to label the UMAP plot clusters. The color in the heatmap is referred to the median of the *arcsinh* marker expression (0–1 scaled) calculated over cells from all the samples. Blue represents lower expression, while red represents higher expression. Light gray bar along the rows (clusters) and values in brackets indicate the relative sizes of the clusters. N naive, T$_{SCM}$ T stem cell memory, CM central memory, TM transitional memory, EM effector memory, EMRA effector memory reexpressing the CD45RA, cTfh circulating T follicular helper cells. **D** UMAP graphs stratified by therapy: healthy donors (HD); Cladribine; Dimethyl Fumarate (DMF); DMF Lymphopenic; Fingolimod; interferon 1β (IFN); Natalizumab; Teriflunomide; Rituximab/Ocrelizumab. **E** Dot plots of different subpopulation of Ag⁺ T cells in patients treated with different DMT. The central bar represents the mean ± SD. Kruskal–Wallis test with Benjamini–Hochberg correction for multiple **A–E**: HD: *n* = 13; Cladribine: *n* = 6; DMF: *n* = 14; DMF Lymphopenic: *n* = 9; Fingolimod: *n* = 12; IFN: *n* = 13; Natalizumab: *n* = 15; Teriflunomide: *n* = 8; Rituximab/Ocrelizumab: *n* = 11.

for the absolute numbers. A detailed representation of all statistical differences among therapies for each cluster is reported in detail in the Supplementary Fig. 3.

### Patients treated with teriflunomide develop higher percentage of Ag⁺ CD8⁺ T cells if compared to healthy donors

Next, we aimed to investigate Ag⁺ CD8⁺ T cells. Figure 2A shows that lymphopenic MS patients treated with DMF showed lower percentage of CD8⁺ T cells if compared to HD. Absolute number of CD8⁺ T cells was lower in cladribine-, DMF- (both groups of patients) or FTY-treated patients if compared to HD. Natalizumab-treated patients displayed the highest absolute number when compared to cladribine-, DMF-, FTY- or IFN-treated patients. Lymphopenic DMF-treated patients showed the lowest absolute number of CD8⁺ T cells when compared to the other treatments. Figure 2B indicates that, as far as the percentage of Ag⁺ CD8⁺ T cells is considered (gating strategy is shown in Supplementary Fig. 4), teriflunomide-treated patients displayed the highest percentage. DMF- or FTY-treated patients showed the lowest absolute number if compared to HD.

Then, we applied the aforementioned unsupervised method of analysis, *i.e.*, FlowSOM, for the identification of the phenotype of Ag⁺ CD8⁺ T cells (Fig. 2C, Supplementary Fig. 5). We could identify 12 different clusters, spanning from that of naïve Tc0 cells to terminally differentiated effector memory T cells re-expressing CD45RA (EMRA), that were also CD57⁺ and/or PD-1⁺. More in detail, FlowSOM revealed one cluster of CM Tc0/2 (CD45RA⁻CCR7⁺CD27⁺CD28⁺PD-1⁺), one cluster of TM Tc1 expressing CXCR5 (CD45RA⁻CCR7⁻CD27⁺CD28⁺PD-1⁺CXCR5⁺CXCR3⁺), five clusters of EM (mainly Tc0/2, Tc17 or Tc1 expressing or not CD57 and PD-1) and four clusters of EMRA (mainly Tc0/2, Tc17 or Tc1 expressing or not CD57 and PD-1). The phenotype of Ag⁺ CD8⁺ T cells of DMT-treated patients was quite different not only from that of HD, but also among different therapies (Fig. 2D). Natalizumab-treated patients displayed the highest percentage of CM Tc0/2 cells, while FTY-treated patients displayed the lowest percentage of TM Tc1 CXCR5+ cells. Natalizumab-treated patients showed the highest percentage and absolute numbers of CM Tc0-2 and TM Tc1 CXCR5⁺, (Fig. 2E). All differences in the phenotype are reported in detail in the Supplementary Fig. 6.

### MS patients treated with different DTM reveal polyfunctional profiles

The functional properties of Ag⁺-specific T cells were investigated by measuring the percentages of cells producing IFN-γ, tumor necrosis factor (TNF), interleukin (IL)−2, IL-17, and/or granzyme B (GRZB), along with the expression of the degranulation marker CD107a. The percentages of cells producing cytokines were assessed after 16 h of in vitro stimulation with a SARS-CoV-2 peptide pool covering the complete sequence of SARS-CoV-2 spike glycoprotein (Supplementary Figs. 7–10).

MS patients treated with FTY showed the highest percentage of CD4⁺ T cells producing GRZB and the lowest percentage of CD4⁺ T cells producing IL-2, displaying a more cytotoxic profile, that is typically found in autoimmune diseases and, in particular, during MS (Fig. 3A). Polyfunctional properties were investigated in CD4⁺ and CD8⁺ T cells by analyzing the simultaneous production of TNF, CD107a, IFN-γ, IL-2, and IL-17 using the bioinformatic Simplified Presentation of Incredibly Complex Evaluation (SPICE) tool. Healthy donors displayed a different polyfunctional profile if compared to FTY-, natalizumab-, teriflunomide- or rituximab/ocrelizumab-treated MS patients. The polyfunctional profile of natalizumab-treated patients was different. These differences were mainly due to the percentage of CD4⁺ T cells producing CD107a⁺IFN-γ⁻IL2⁺IL17⁻TNF⁺, CD107a⁻IFN-γ⁺IL2⁺IL17⁻TNF⁺ and CD107a⁻IFN-γ⁻IL2⁺IL17⁻TNF⁺ (Fig. 3B).

Regarding Ag⁺ CD8⁺ T cell, FTY-treated patients showed a higher percentage of CD8⁺ T cells producing GRZB if compared to those treated with DMF, natalizumab or IFN (Fig. 3C). The polyfunctional profile of CD8⁺ T cells of HD was then different from those of rituximab/ocrelizumab-treated patients. The percentage of CD8⁺ T cells that were CD107a⁻IFN-γ⁻IL2⁻IL17⁻TNF⁺ was higher in teriflunomide-treated patients if compared to those treated with DMF (Fig. 3D).

### Fingolimod- or rituximab/ocrelizumab-treated-patients displayed low or undetectable levels of Ag⁺ B cells

Natalizumab-treated MS patients were characterized by highest percentage and absolute number of B cells if compared to all groups (Fig. 4A). Consistent with expectations, patients treated with rituximab/ocrelizumab exhibited markedly reduced, but detectable levels of circulating B cells. Subsequently, the percentage of Ag⁺ B cells was quantified, revealing that rituximab/ocrelizumab-treated patients displayed the lowest proportion and absolute count of these cells (Fig. 4B, Supplementary Fig. 11). Furthermore, patients treated with FTY showed a reduced percentage and absolute count of these cells compared to HD. Moreover, the phenotype of Ag⁺ B cells was extensively characterized using the aforementioned unsupervised methods. (Supplementary Fig. 12). Ag⁺ B cells were composed by 11 clusters, such as: naïve (CD20⁺CD21⁺CD24⁺CD38⁻IgD⁺IgM⁺); transitional B cells (TrB; CD20⁺CD21⁺CD27⁻CD24⁺CD38⁺IgD⁺IgM⁺); immature TrB (CD20⁺CD21⁻CD24⁺CD27⁺CD38⁺IgD⁺IgM⁺); six clusters of memory B cell MBCs defined as follows: MBC unswitched (CD20⁺CD21⁺CD24⁺CD27⁺IgD⁺IgM⁺), MBC IgA⁺(CD20⁺CD21⁺CD24⁺CD27⁺IgA⁺IgD⁻IgM⁻), MBC IgG⁺ CD21low (CD20⁺CD21$^{low}$CD24⁺CD27⁺IgG⁺), MBC IgG⁺ CD20⁻ (CD21⁺ CD24⁺CD27⁺), and MBC IgG⁺ CD71⁺ (CD20⁺CD21⁺CD24⁺CD27⁺IgG⁺) and MBC IgG⁺ (CD20⁺CD21⁺CD24⁺CD27⁺CD71⁻IgG⁺); plasmablasts (PB) were defined as PB CD27⁺CD71⁺CD38⁺⁺; atypical B cells (atBCs) as CD21⁻CD27⁻CD20⁺IgG⁺ (Fig. 4C). Ag⁺ B cells were phenotypically very similar in all groups except for rituximab/ocrelizumab treated patients (Fig. 4D and Supplementary Fig. 13). The highest percentage of naïve Ag⁺ B cells was found in FTY-treated MS patients and the lowest in

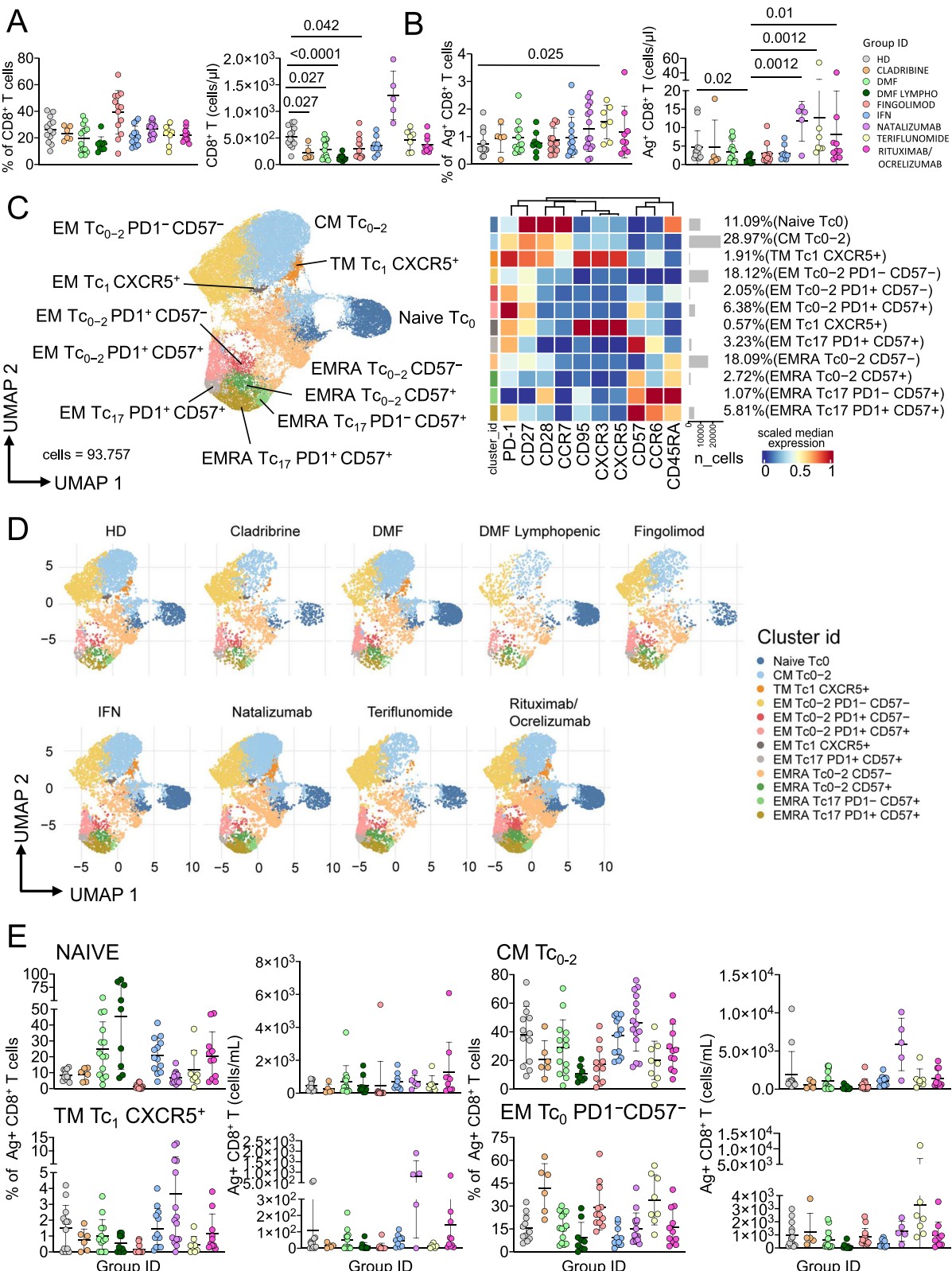

rituximab/ocrelizumab-treated patients, and they both displayed the lowest absolute number of this cell population. However, natalizumab-treated patients displayed the highest absolute number. Moreover FTY-treated patients had the lowest percentage and absolute number of Ag$^+$ MBC IgA cells.

Finally, the plasma levels of anti-spike IgG were measured, and nearly all MS patients developed humoral immunity. However, among

the patients treated with FTY, 1 out of 14, and among those treated with rituximab/ocrelizumab, 3 out of 11, did not develop IgG. These groups, along with cladribine-treated patients, showed the lowest IgG concentrations. Additionally, when analyzing the levels of neutralizing antibodies (anti-RBD), 6 out of 8 rituximab/ocrelizumab-treated patients, 11 out of 13 FTY-treated patients, and 5 out of 6 cladribine-treated patients exhibited positive neutralizing capacity (Fig. 4E).

**Fig. 2 | Ag⁺ CD8⁺ T cell landscape. A** Percentage and absolute number of CD8⁺ T cells. Dot plots show the percentage and absolute number of CD8⁺ T cells. The central bar represents the mean ± SD. Kruskal−Wallis test (one-sided) with Benjamini−Hochberg correction for multiple comparisons. Significant adjusted q-values are reported in the figure. **B** Percentage and absolute number of Ag⁺ CD8⁺ T cells. Dot plots show the percentage and absolute number of Ag⁺ CD8⁺ T cells. The central bar represents the mean ± SD. Kruskal−Wallis test (one-sided) with Benjamini−Hochberg correction for multiple comparisons. Significant adjusted q-values are reported in the figure. **C** Ag⁺ CD8⁺ T cells phenotype UMAP and Heatmap. Uniform Manifold Approximation and Projection (UMAP) plot shows the 2D spatial distribution of 93.757 cells from healthy donors (HD) and MS patients treated with different DMT embedded with FlowSOM clusters. Heatmap of the median marker intensities of the 10 lineage markers across the 12 cell populations obtained with FlowSOM algorithm after the manual metacluster merging. The colors of cluster_id column correspond to the colors used to label the UMAP plot clusters. The color in the heatmap is referred to the median of the *arcsinh* marker expression (0−1 scaled) calculated over cells from all of the samples. Blue represents lower expression, while red represents higher expression. Light gray bar along the rows (clusters) and values in brackets indicate the relative sizes of the clusters. N naive, T_SCM T stem cell memory, CM central memory, TM transitional memory, EM effector memory, EMRA effector memory reexpressing the CD45RA, cTfh circulating T follicular helper cells. **D** UMAP graphs stratified by therapy: healthy donors (HD); Cladribine; Dimethyl Fumarate (DMF); DMF Lymphopenic; Fingolimod; interferon 1β (IFN); Natalizumab; Teriflunomide; Rituximab/Ocrelizumab. **E** Dot plots of different subpopulation of Ag⁺ T cells in patients treated with different DMT. The central bar represents the mean ± SD. Kruskal−Wallis test with Benjamini−Hochberg correction for multiple. In **A**–**E** plots: HD: n = 13; Cladribine: n = 6; DMF: n = 14; DMF Lymphopenic: n = 9; Fingolimod: n = 12; IFN: n = 13; Natalizumab: n = 15; Teriflunomide: n = 8; Rituximab/Ocrelizumab: n = 11.

## Ag⁺ T cells from MS patients treated with different DMT switch on different metabolic features

After in vitro stimulation with the SARS-CoV-2 peptide pool, the metabolism of cells expressing CD69 and CD137 within CD4⁺ and CD8⁺ T cells was investigated by using single-cell metabolic regulome profiling (scMEP), a technique that quantifies proteins that regulate metabolic pathway activity by 45-parameter mass cytometry [adapted from ref. [36]]. PBMC from patients treated with different DMT were stained with 22 mAbs recognizing cell phenotype and 23 mAbs recognizing molecules involved in different metabolic pathways (see Supplementary Table 5) and analyzed by mass cytometry. We identified all major cell lineages of Ag⁺ T cells (Supplementary Figs. 14−16), and then we focused on their metabolic states (Fig. 5).

Along with their distinct functions, Ag⁺ T cell lineages also possess unique metabolic profiles which are essential for their function and maintenance. T cell activation is indeed accompanied by a switch from a metabolism mainly based upon mitochondrial respiration to a metabolism where the glycolytic flux is prevalent[37].

Clustering cells on the basis of the expression of proteins involved in different metabolic pathways (*i.e.*, GLUT1, MCT1, GAPDH, LDHA, HK2, PFKB4, G6PD, CytC, CS, IDH1, ATPA5, CD98, GLUD1/2, CD36, CPT1A, VDAC1, pACC, pPGC1a, pS6, pPDK1, HIF1a, pNRF2, pH3), we identified 10 scMEP states (Supplementary Figs. 17−19). These metabolic states spanned from cells with a quiescent or exhausted metabolism (scMEP1, 2, 3) to those with high activation of different metabolic pathways (scMEP6, 9, 10) (Fig. 5A−C). In particular, 20% of Ag⁺ T cells were characterized by the scMEP1 state, a basal level of metabolic activation (low expression of all metabolic features, except from MCT1 and CS), while 44% of Ag⁺ T cells was grouped into scMEP2 state, a metabolic quiescent/exhausted state (with very low levels of all markers except PFKB4 and CPT1A). Cells in scMEP3 state displayed high levels of GLUD1/2 and CD36, meaning that amino acid metabolism as well as fatty acid oxidation (FAO) were activated; scMEP4 state described a metabolic phenotype of pentose pathway activation (high expression of G6PD), activation of tricarboxylic acid (TCA) cycle (IDH1, ATP5A), involvement of amino acid pathway (high expression of GLUD1/2) and enhanced mitochondrial dynamics (phosphorylation of VDAC1). scMEP5 and scMEP7 states grouped cells characterized by low activation of mainly glycolysis, pentose, oxidative phosphorylation (OXPHOS) and FAO (high expression of MCT1, HK2, PFKB4, CytC, CS, CD98, GLUD1/2). A total of 20% of Ag⁺ T cells was grouped into scMEP6 (4.96%) and scMEP9 (14.88%) states, characterized by a glycolytic profile (high protein expression of HK2 and PFK4) and cell growth (pPDK1). scMEP8 represented 0.33% of Ag⁺ T cells, characterized by glycolytic activation (high expression of GLUT1, MCT1, HK2) as well as activation of amino acid pathway (CD98) and mTOR activation (pS6). Finally, 0.46% of Ag⁺ T cells clustered into scMEP10, a highly activated metabolic state where all metabolic pathways taken into account are switched on.

As far as different therapies were considered, the distribution of Ag⁺ T cell on the basis of metabolic states was different (Fig. 5D). In particular, FTY-treated patients were characterized by the lowest percentages of Ag⁺ T cells in scMEP4 and scMEP9 states, meaning that these cells do not rely on glycolysis and pentose pathway. When compared to HD, Ag⁺ T cells from DMF-treated patients (the non-lymphopenic ones) and IFN-treated ones displayed a low proportion of scMEP2, suggesting that the drug is able to reprogram metabolism.

Because Ag⁺ T cell scMEP states were defined using exclusively metabolic features, we tried to associate their metabolic features with phenotypic properties (Fig. 5E). As expected, scMEP metabolic states were clearly linked to immunological phenotypes. scMEP2 state was composed by terminally differentiated (EMRA) CD8⁺ T cells expressing low level of CD27, high level of CD57 and medium level of PD-1. scMEP4 state was mainly represented by effector memory CD4⁺ T cells expressing CD27, while scMEP 9 state was composed by a small cluster of cycling. We found that a few cells were present in scMEP10, which is formed by activated (HLA-DR⁺CD38⁺) Ag⁺ CD8^dim T cells expressing CD57 and CD11c; such population is typically present in autoimmune diseases, is expanded in an Ag-dependent manner and mainly produce IFN-γ[38].

To determine the pattern of the dynamic development of scMEP states, trajectory inference analysis has been applied (Fig. 5F). After in vitro stimulation, Ag⁺ T cells acquire different metabolic states starting from scMEP5, passing through highly metabolic activated state (scMEP5, 7 and 9), and reaching the final state of metabolic quiescence (scMEP1, 2 and 3).

Finally, on freshly isolated PBMC from a small subgroup of patients, we could profile the global metabolic capacities and dependencies of Ag⁺ T cells by the SCENITH assay[39]. We found that Ag⁺ CD4⁺ T cells from IFN-treated patients displayed a higher glucose dependence when compared to HD, while natalizumab-treated patients displayed the highest glycolytic capacity (Supplementary Fig. 20).

## Ag⁺ plasmablasts from FTY-treated MS patients are fully glycolytic while those from the rituximab/ocrelizumab group display a quiescent/senescent metabolism

Metabolic demands of Ag⁺ B cells are related to their functional activity. Upon stimulation, Ag⁺ B cells have a balanced increase in lactate production and oxygen consumption, with proportionally increased GLUT1 expression leading to enhance glucose uptake and mitochondrial mass[40]. Moreover, their proliferation and function are regulated by HIF-1α expression. Here, we found that Ag⁺ B cells displayed 8 different metabolic states (Fig. 6A−C and Supplementary Figs. 21−23). Approximately 5.26% of cells were classified into scMEP1, a metabolic state marked by the upregulation of GLUT1. Another 16.91% exhibited activation in TCA, ECT, signaling, and transcription, categorizing them as scMEP2. The majority, comprising over 60% of cells, fell into scMEP3, scMEP4, and scMEP6 groups. These states displayed

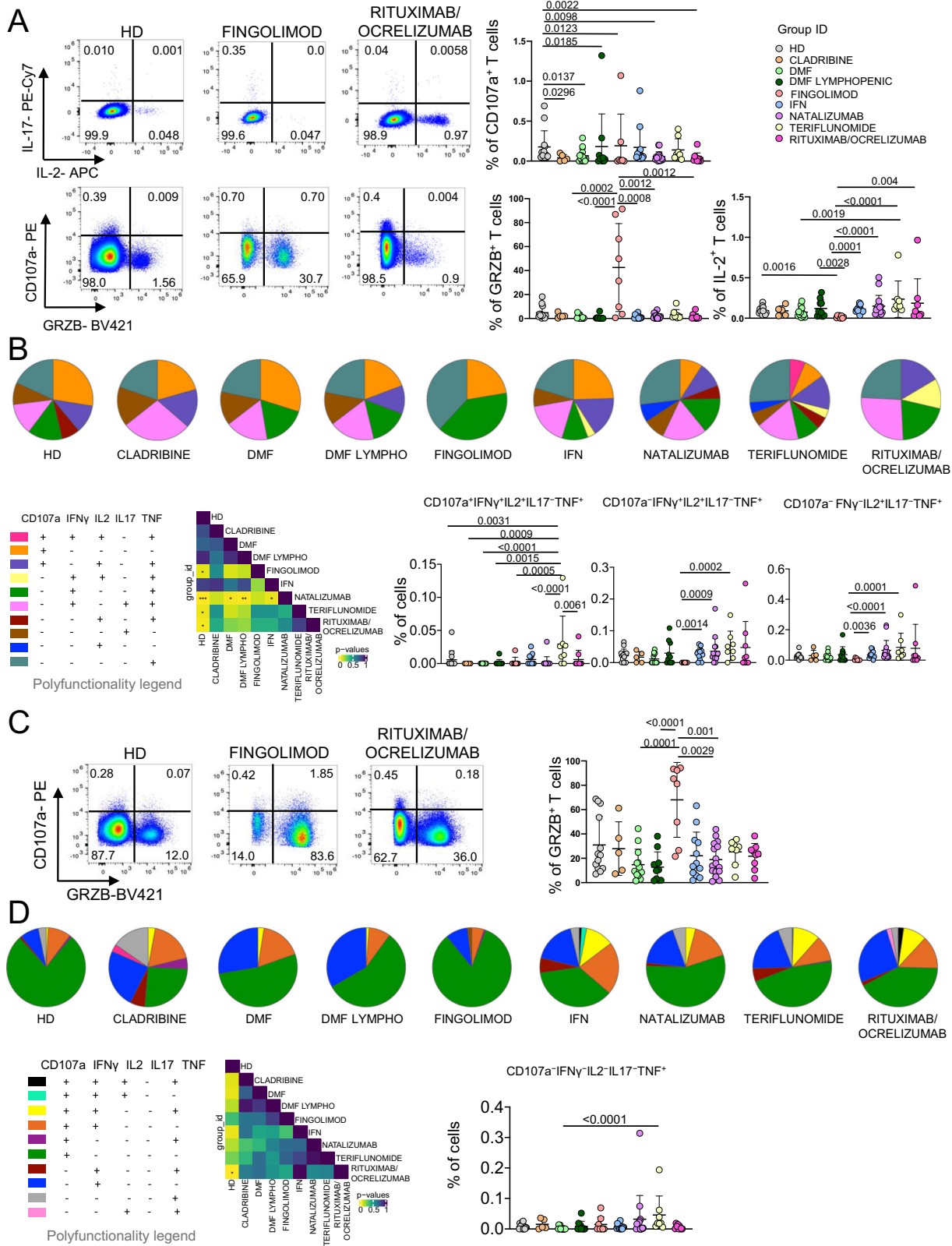

similar metabolic activities, excluding glycolysis. Notably, cells in the scMEP4 state exhibited elevated CD36 expression, indicating heightened activity in the fatty acid pathway. About 6.42% of Ag+ B cells clustered into scMEP5, characterized by reduced glycolytic activation but increased engagement in the pentose pathway, ETC, TCA, fatty acid metabolism, mitochondrial dynamics, proliferation, and signal transduction. Additionally, less than 2% of non-proliferating cells were

distributed into scMEP7 and scMEP8 states. These states exhibited low mTOR and pentose pathway activation but demonstrated glycolytic, TCA, ECT activation.

The distribution of metabolic states changed significantly across different groups, including HD and MS patients treated with DMTs (Fig. 6D). In particular, patients treated with FTY and rituximab/ocrelizumab exhibited the highest proportion of Ag+ B cells in scMEP1 state.

**Fig. 3 | Ag⁺ CD4⁺ and CD8⁺ T cell functionality. A** Percentage of Ag⁺ CD4⁺ T cells producing different cytokines after in vitro stimulation with SARS-CoV-2 peptides. Representative dot plots showing the percentages of CD4⁺ Ag⁺ cells producing IL-2, IL-17, CD107a and granzyme B (GRZB). Plots show mean (center bar) ± SD. Kruskal–Wallis (one-sided) test with Benjamini-Hochberg correction for multiple comparisons. **B** Polyfunctional profile of Ag⁺ CD4⁺ T cells. (Upper) Pie charts representing the proportion of Ag⁺ CD4⁺ T cells producing different combinations of CD107a, IL2, IL17, IFNγ, and TNF. Each color refers to specific polyfunctional CD4 T subpopulation as reported in the 'polyfunctionality legend'. The far-left heatmap illustrates the statistical variances among the 9 distinct pie charts; The central bar represents the mean ± SD. Kruskal–Wallis (one-sided) test with Benjamini-Hochberg correction for multiple comparisons. (far-right) Dot plot reporting the percentages of Ag⁺ CD4⁺ producing different combination of cytokines. Kruskal-Wallis (one-sided) test with Benjamini–Hochberg correction for multiple comparisons. **C** Percentage of Ag⁺ CD8⁺ T cells producing different cytokines after in vitro stimulation with SARS-CoV-2 peptides. Representative dot plots of Ag⁺ CD8⁺ cells producing CD107a and GRZB. Dot plot representing the percentage of Ag⁺ CD8⁺ T cells producing GRZB is shown, mean (center bar) ± SD. Kruskal–Wallis test (one-sided) with Benjamini–Hochberg correction for multiple comparisons. **D** Polyfunctional profile of Ag⁺ CD8⁺ T cells. (Upper) Pie charts representing the percentage of Ag⁺ CD8⁺ T cells producing different combinations of CD107a, IL2, IL17, IFNγ, and TNF. Each color refers to specific polyfunctional CD8 T subpopulation as reported in the 'polyfunctionality legend'. The far-left heatmap illustrates the statistical variances among the 9 distinct pie charts; Kruskal-Wallis test (one-sided) with Benjamini–Hochberg correction for multiple comparisons (Right) Dot plot reporting the percentage of Ag⁺ CD8⁺ CD107a⁻IFNγ⁻IL2⁻IL17⁻TNF population. The central bar represents the mean ± SEM. Kruskal–Wallis test with Benjamini–Hochberg correction for multiple comparisons was used to test the differences among the nine groups. In **A–C** plots: HD healthy donors (N = 13); Cladribine (N = 6); DMF Dimethyl Fumarate (N = 14); DMF Lymphopenic: Dimethyl Fumarate Lymphopenic (N = 9); Fingolimod (N = 12); IFN Interferon 1β (N = 13); Natalizumab (N = 15); Teriflunomide (N = 8); rituximab/ocrelizumab (N = 11).

However, they showed a lower percentage of cells in scMEP3 compared to patients treated with natalizumab. Those treated with rituximab/ocrelizumab displayed the lowest percentage of cells in scMEP4. As far as the phenotype of these B cells is concerned, plasmablasts constituted most of cells in scMEP1, scMEP7 and scMEP8; memory B cells were grouped in scMEP3 and scMEP4; recently activated Ag⁺ B cells were in scMEP2 while atypical B cells (atBC) were in scMEP5 (Fig. 6E). These results are in line with the clustering performed by using both lineage and metabolic markers (Supplementary Figs. 24–25).

As for Ag⁺ T cell, we determined the dynamic development of scMEP states by using trajectory inference analysis (Fig. 6F). After in vitro stimulation, Ag⁺ B cells acquired different metabolic states starting from scMEP2, passing through mid-metabolic activated state (scMEP3, 4 and 5), reaching the final state of highly metabolic activation (scMEP7, 8 and 1).

Finally, due to limited cell number, we could use the SCENITH assay on freshly isolated PBMC only from a limited number of patients treated with different DMT. We found that Ag⁺ B cells from natalizumab-treated patients displayed a trend of higher glycolytic capacity when compared to HD (Supplementary Fig. 26). Even the results are not statistically significant, this observed trend seems to confirm what we have found by the scMEP.

### FTY- and natalizumab-treated patients develop a different antigen-specific immune response

In order to describe an immunological signature indicating how different DMT could shape Ag-specific immunity and protection against SARS-CoV-2, we took advantage of the use of the principal component analysis (PCA)[7]. Based on the first two PCs, PCA revealed that only FTY- and natalizumab-treated patients develop a clearly different quality of the Ag-specific immune response (p < 0.05, Fig. 7A), forming separate clusters, while all other groups (also including the rituximab/ocrelizumab one) displayed similar immunological features even when compared to HD.

As shown in Fig. 7A, FTY-treated patients form a cluster on the left side of PC1 (whose weight was 13.9%) while natalizumab-treated patients are on the right side. Figure 7B reveals that the main responsible of this division were the immunological features more represented in FTY-treated patients, such as: Ag⁺ cytotoxic CD4⁺ and CD8⁺ T cells (expressing CD107a and Granzyme), cells in scMEP2 (quiescent metabolic state), Ag⁺ T cells expressing of CD57 and PD-1 (indicating senescence and exhaustion/activation). On the contrary, the main features responsible for the clusterization of natalizumab-treated patients were the absolute number of Ag⁺ B and Ag⁺ CD4⁺ T cells, the number of B and CD8⁺ T cells, the marked shift of Ag⁺ T cell towards Th1 phenotype, and more marked metabolic status in B cells (scMEP3).

To better point out the distinctive features of the immune response in patients undergoing treatment with natalizumab or FTY, we employed PCA specifically on these patient groups (Fig. 7C, left panel). FTY patients are divided from patients treated with natalizumab according to PC1 (26.5%). Besides the aforementioned immunological features responsible for the different clusterization, also that different metabolic engagement by Ag⁺ B cells was responsible for the division (Fig. 7C, right panel). Indeed, FTY-treated patients displayed higher percentage of glycolytic Ag⁺ B cells in scMEP1 and scMEP7 states together with higher percentage of Ag⁺ T cells in scMEP2, characterized by low level of all metabolic markers, except from PFKB4 and CPT1A.

### High percentage of Ag⁺ T cells in scMEP 10 and scMEP7 as well as a high percentage of Ag⁺ B cells in scMEP 1, scMEP2 and scMEP5 predict protection from SARS-CoV2-breakthrough infection

Accurate identification of phenotype-relevant subsets from heterogeneous cell populations is crucial to delineate an immunological signature that could predict protection from breakthrough infections. To pursue this, and to identify subpopulations that could be associated with categorical or continuous phenotypes from single-cell data, we used a supervised learning framework called "phenotype-associated subpopulations from single-cell data" (PENCIL), based on rejection strategy learning[41]. Using this classification mode, we interrogated subpopulations of Ag⁺ T and B cells that were associated with individuals that experienced symptomatic SARS-CoV-2 infection after the third dose of vaccine.

A total of 18 individuals (MS patients and HD, as reported in Supplementary Data 1) experienced SARS-CoV-2 infection within 6 months from the last dose of vaccine. Taking into consideration data from Ag⁺ T cells analyzed with scMEP, we performed the prediction analysis (see Methods). PENCIL revealed that high percentages of Ag⁺ T cells grouped into scMEP7 (effector memory CD4⁺CD27⁺PD-1dim CD57dim) and scMEP10 (mainly formed by CD8dimCD11c⁺ T cells with almost all metabolic pathways activated and by T cell, also expressing PD-1 and CD57) were associated with absence of SARS-CoV-2 infection, meaning that these clusters of cells can confer high immune protection. On the contrary, a high percentage of cells grouped into scMEP9 (CD4⁺ TM) and scMEP2 (either CD4 or CD8, either EM or EMRA, expressing PD-1 and CD57 with a metabolically resting phenotype) were associated with the onset of a SARS-CoV-2 breakthrough infection (Fig. 8A, B). To validate this prediction, patients and HD were stratified by therapy and we observed that high percentage of scMEP2 (in the case of HD and IFN-treated individuals) and scMEP9 (in rituximab/ocrelizumab group) were associated with breakthrough infection (Supplementary Fig. 27).

Then, Ag⁺ B cells that underwent scMEP analysis were interrogated by PENCIL (see Methods). PENCIL indicated that a high percentage of scMEP1 (not proliferating plasmablasts, highly glycolytic),

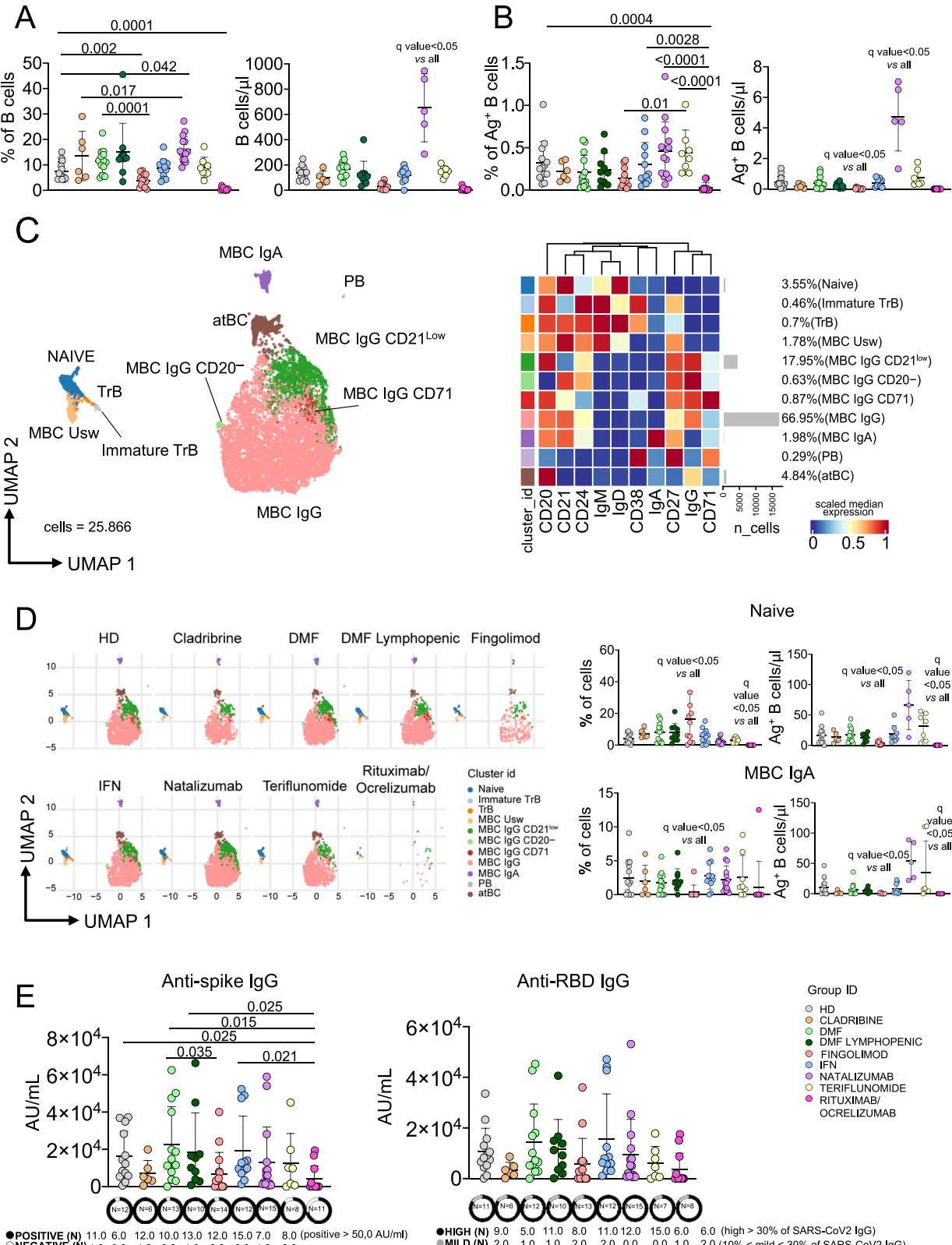

scMEP2 cells (i.e., recently activated naïve B cells with activated mitochondrial metabolism), and scMEP5 cells (atypical B cells, coupled with activation of all metabolic pathways) predict protection from SARS-CoV-2 infection. On the contrary, a high percentage of cells grouped into scMEP3 (memory B cells, PD-1 dim) predict SARS-CoV-2 infection (Fig. 8C, D). We validate this prediction as performed with Ag+ T cells. Even if not significant for the low number of cells and patients, high percentage of scMEP3 (as in the case of HD, Natalizumab and rituximab/ocrelizumab) can predict breakthrough infection (Supplementary Fig. 27).

## Discussion

The aim of this study was to ascertain whether COVID-19 vaccinated patients affected by the relapsing-remitting form of multiple sclerosis

**Fig. 4 | Ag⁺ B cell landscape. A** Dot plot shows the total percentage and the absolute number of CD19⁺ B cells. The central bar represents the mean ± SD. Kruskal–Wallis test (one-sided) with Benjamini–Hochberg correction for multiple comparisons. **B** Dot plot shows the percentage and absolute number of antigen-specific CD19⁺ B cells. The central bar represents the mean ± SD. Kruskal–Wallis test (one-sided) with Benjamini–Hochberg correction for multiple comparisons was used to test the differences among the nine groups. **C** UMAP plot shows the 2D spatial distribution of 25.866 antigen-specific B cells from healthy controls (HD) and patients with Multiple Sclerosis embedded with FlowSOM clusters. Heatmap of the median marker intensities of the 10 lineage markers across the 11 cell populations obtained with FlowSOM algorithm after the manual metacluster merging. The colors of cluster_id column on the left correspond to the colors used to label the UMAP plot clusters. Each color in the heatmap is referred to the median of the arcsinh marker expression (0–1 scaled) calculated over cells from samples. Blue represents lower expression. while red represents higher expression. Light gray

histogram bar and values indicate the relative sizes of the clusters. Naive; TrB. transitional B cells; MBC Usw. memory B cell unswitched; MBC memory B cell, PB plasmablasts, atBC atypical B cell. **D** (Left) UMAP graphs stratified by therapy. (Right) Dot plot showing the percentages and absolute numbers of naïve and MBC IgA B cells. The central bar represents the mean ± SD. Kruskal–Wallis test (one-sided) with Benjamini–Hochberg correction for multiple comparisons was used to test the differences among the nine groups. **E** Anti-spike and anti-RBD IgG concentrations in plasma samples from HD and MS treated groups. The central bar represents the mean ± SD. Kruskal–Wallis test (one-sided) with Benjamini–Hochberg correction for multiple comparisons. Adjusted *P*-values are indicated in the figure. Plots **A**–**E** HD healthy controls (*N* = 13); Cladribine (*N* = 6); DMF: Dimethyl Fumarate (*N* = 14); DMF Lymphopenic: Dimethyl Fumarate Lymphopenic (*N* = 9); Fingolimod (*N* = 12); IFN: Interferon 1β (*N* = 13); Natalizumab (*N* = 15); Teriflunomide (*N* = 8); Rituximab/Ocrelizumab (*N* = 11).

and receiving different MS-treatments would mount an effective T and B cell response against SARS-CoV-2. For this reason, we have used different approaches and techniques mainly based on flow and mass cytometry, to carefully investigated cell phenotype, function and metabolism.

Several observational studies evaluating the effectiveness of COVID-19 vaccines in MS patients treated with DMT showed that most of these drugs allow for mounting a protective immune response, at least in terms of antibody production and production of antigen-specific B and T cells, even if some patients can experience a reduced immune response. However, an immune signature associated with the phenotype and function of Ag⁺ T and B cells that could suggest the existence of a predisposition to breakthrough infection in MS patients has never been investigated.

Here, we show that nearly 6 months after the third SARS-CoV-2 vaccine dose, the overall SARS-CoV-2-specific T and B cell response in relapsing-remitting MS patients treated with different drugs was similar among all patients and healthy donors, except for those treated with FTY or natalizumab, whose cells displayed totally different immunological features as well as a diverse immunometabolic engagement.

In the case of FTY-treated patients, we saw that phenotype, function, and metabolism of Ag⁺ T and B cells seemed to mimic the characteristic of an aged immune system. Indeed, these patients were characterized by high proportions of effector memory T cells expressing PD-1 and CD57; CD4⁺ T cells producing granzyme; Ag⁺ T cells with low polyfunctional profile; decreased percentages of Ag⁺ B cells. Moreover, most Ag⁺ T and B cells display a glycolytic metabolism which might constitute a rescue mechanism to maintain an activated and functional phenotype when metabolism is skewed due to the possible, well known age-dependent mitochondrial impairment[42,43].

The immune system of MS patients is characterized by a premature aging[44], and DMT can cause drastic changes that worsen or even accelerate immune senescence long after the drug has been stopped[45]. The effects of cell depleting agents are not readily reversible, and even those of therapies primarily targeting cell migration such as natalizumab and FTY may long lasting effects. Aging of the immune system involves not only a decreased production of naïve T cells, but also an increase in terminally differentiated late effector memory T cells determining a narrowing of the T cell repertoire[46,47], with an increase in the level of activation and cytotoxicity[8]. Aging decreases B cell differentiation in the bone marrow and the output of mature B cells, induces a redistribution of B cell subsets in the periphery with a significant increase in frequencies and numbers of proinflammatory B cells, decreases the expression of molecules involved in Ig class-switch recombination and somatic hypermutation, two processes leading to the generation of high-affinity protective antibodies as well as germinal center formation, and decreases B cell repertoire diversity[5,48]. These phenotypic and functional alterations

are strictly connected to age-associated metabolic changes that impact the bioenergetic program of T and B cells at discrete phases of development and activation[49]. Glycolysis and LDH activity are reduced in aged T cells[50] and mitochondrial dysfunction is one of the main hallmarks of aged T cells, that are characterized by smaller or dysfunctional mitochondria characterized by a decreased respiration rate and ATP production[37].

We found that MS patients treated with different DMT display a different phenotype of Ag⁺ T and B cells. It has been shown that some effects of FTY, DMF and rituximab/ocrelizumab resemble immunosenescence, as they cause a decrease in total B and T cells and induce a negative regulation of Th1 and Th17 differentiation while promoting Th2 differentiation[51]. FTY not only modulates lymphocyte trafficking, but also modulates the composition of B and T cells subsets, with an increase of circulating effector memory T cells and decrease of naïve T cells. On the contrary, natalizumab induces an increase in total T cells (including Th1 and Th17), total B cells, memory B cells, but alter the proportion of plasmablasts which have high expression of CD49d[45,52]. Moreover, most Ag⁺ B cells of natalizumab-treated patients were metabolically quiescent. Natalizumab binds CD49d (integrin α4) which is also a molecule expressed during cell activation[53]. Given that natalizumab prevents this phenomenon, likely it also prevents metabolic activation of B cells and their capability to differentiate and produce antibodies. The metabolic pathways within immune cells regulate the formation of antigen-specific immune and its cell function. Indeed, metabolic pathways can influence the development of various T helper subsets. For example, Treg cells predominantly depend on OXPHOS and mitochondrial FAO for development and survival, whereas the generation of Th17 cells requires glycolysis[54]. Here, for the first time, by applying a method based upon mass cytometry, we confirm how different therapies are able to modify the metabolic profiles of circulating antigen-specific cells.

Assessing the molecular and cellular state of the immune system after vaccination, by adopting data-driven models, could be used to predict pathogen-specific immune responses or the prevention of breakthrough infection. The goal is to identify key immune signatures that are responsible for the creation of an effective immune response. Systems-biology analyses of influenza virus vaccination have identified antibody response predictors, these have been based on post-vaccination parameters, such as the magnitude of plasmablast increases on day 7, and changes in blood host-derived transcripts on days 1–3 after vaccination[55,56]. Moreover, certain immune signatures can predict not only the response to malaria vaccination, but the clinical outcomes of acute infection[57,58]. Then, we applied for the first time to mass cytometry data a prediction approach like PENCIL, that is typically used to analyze single cell transcriptome. We found that predominance of metabolically active Ag⁺ CD4⁺,CD27⁺,PD-1ᵈⁱᵐ CD57ᵈⁱᵐ T cells and CD8ᵈⁱᵐ,CD11c⁺ Ag⁺ T cells correlates with the absence of breakthrough infection and may thus confer high immune protection.

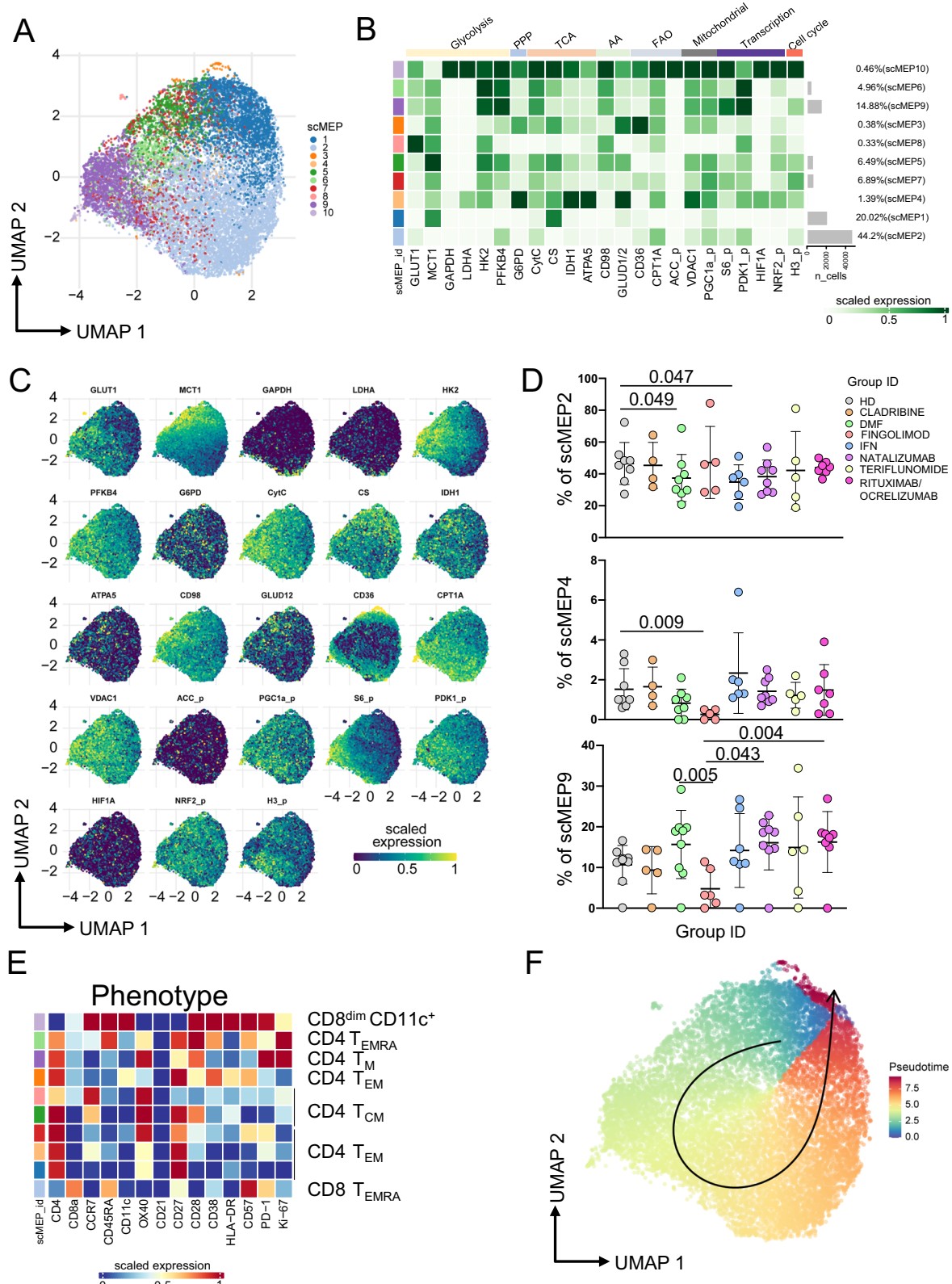

In addition, high percentage of non-proliferating, Ki67⁻ plasmablasts, highly glycolytic, recently activated naïve B cells with activated mitochondrial metabolism and metabolically activated atypical B cells predict protection from SARS-CoV-2 infection. This may indicate these cells are able to recognize antigens but do not to properly react and mount a specific response, paving the way to approaches to increase the effectiveness of vaccination.

Questions regarding how much the immune signature before vaccination influence the creation of a protective immune response needs to be elucidated and how much these immunological signatures are similar across different populations (young, elderly, pregnant, different ethnicities) need to be investigated.

The study has some limitations. First, the number of patients per group is relatively small, and the investigation is cross-sectional.

**Fig. 5 | Ag+ T cell metabolic states. A** Uniform Manifold Approximation and Projection (UMAP) plot shows the 2D spatial distribution of 107,522 cells from HD and MS patients. UMAP dimensionality reduction is calculated using sampled data from all cells and all available metabolic features; through FlowSOM clustering are identified 10 different clusters, defined as scMEP. Cells are colored by their scMEP state. **B** Heatmap of the median marker intensities of the 23 metabolic markers across the 10 cell populations, obtained with FlowSOM algorithm. The colors of cluster_id column correspond to the colors used to label the UMAP plot clusters/ scMEP. The color in the heatmap is referred to the median of the arcsinh marker expression (0–1 scaled): white represents a lower expression, while dark green represents a higher expression. Light gray bar along the rows (clusters/scMEP) and values in percentages indicate the relative sizes of the clusters. **C** Projection of UMAP graphs stratified by HD and all different MS patients. **D** Representative dot plots showing percentages of scMEP2, scMEP4 and scMEP9 among different Group_IDs. The central bar represents the mean ± SD. Kruskal–Wallis test (one-sided) with Benjamini–Hochberg correction for multiple comparisons is used to test the differences among groups. Adjusted q-values are reported in the figure, if significant. healthy donors (HD, $n = 8$), multiple sclerosis patients treated with Cladribine ($n = 4$), Dimethyl Fumarate (DMF, $n = 8$), Fingolimod ($n = 5$), interferon 1β (IFN, $n = 6$), Natalizumab ($n = 8$), Teriflunomide ($n = 5$), Rituximab/Ocrelizumab ($n = 7$). **E** Heatmap of 14 immunological markers enrichment modeling (not used for metabolic clustering) across different scMEP states, showing the relationship between metabolic states and functional properties. Light gray bar along the rows (clusters/scMEP) and values in percentages indicate the relative sizes of the clusters/scMEP. **F** Pseudotime visualization of scMEP development based on the estimated trajectory and envisaged in UMAP space.

However, considering the parameters that we have investigated, all groups were homogeneous and there were no outliers. Second, given that the exposure to SARS-CoV-2 was not controlled, the results from prediction analysis should therefore be considered preliminary and subject to further validation. However, our findings suggest that only FTY and natalizumab modify significantly (in terms of phenotype and metabolic status) the SARS-CoV-2-specific B and T cell composition after vaccination.

## Methods

### Patient's selection
Demographic and clinical characteristics of 93 MS patients and 13 healthy donors (HD), the type of DMT at the time of vaccination, the type of third dose vaccine and median range of time to last administration, prior COVID-19 infection status, and relevant comorbidities are shown in Supplementary Data 1. Patients were eligible for inclusion if they met the following criteria: (a) a confirmed diagnosis of Relapsing-Remitting Multiple Sclerosis (RRMS), and (b) a history of treatment with FTY, dimethyl fumarate, natalizumab, or teriflunomide for a minimum of six months, or having undergone at least two infusion cycles with rituximab or ocrelizumab or completed at least one full cycle of cladribine. Patients on ocrelizumab or rituximab, as per routine clinical practice, underwent SARS-CoV-2 vaccination at least 6 weeks before subsequent infusion or at least 3 months after the last infusion. Exclusion criteria comprised treatment with steroids during the preceding six weeks and a history of COVID-19 before vaccination.

### Blood collection and isolation of mononuclear cells
Up to 30 mL of blood were collected from each patient in vacuettes containing ethylenediamine-tetraacetic acid (EDTA). Blood was immediately processed. Isolation of peripheral blood mononuclear cells (PBMC) was performed using ficoll-hypaque according to standard procedures. For all experiments, except those related to metabolic investigation, PBMC were stored in liquid nitrogen in fetal bovine serum (FBS) supplemented with 10% dimethyl sulfoxide (DMSO). For metabolic investigation, PBMC were used immediately after isolation. Plasma was stored at −80 °C until use. The study was reviewed and approved by each participant, including healthy donors, provided informed consent according to Helsinki Declaration, and all uses of human material have been approved by the local Ethical Committee (Comitato Etico dell'Area Vasta Emilia Nord, protocol number 199/ 2022, May 24th, 2020) and by the University Hospital Committee (Direzione Sanitaria dell'Azienda Ospedaliero Universitaria di Modena, protocol number 5974, February 24th, 2023). The patients/participants provided their written informed consent to participate in this study.

### Activation induced cell marker assay (AIM) and T cell phenotype
Isolated PBMCs were thawed and rested for 6 h. After resting, CD40-blocking antibody (0.5 mg/ml final concentration) (Miltenyi Biotec,

Bergisch Gladbach, Germany) was added to the cultures 15 min before stimulation. PBMCs were cultured in 96-well plate in the presence of 15-mer peptides with 11-amino acids overlap, covering the complete sequence of Wuhan SARS-CoV-2 Spike glycoprotein (PepTivator SARS-CoV-2 Prot_S complete, Miltenyi Biotec, Bergisch Gladbach, Germany) together with 1 µg/mL of anti-CD28 (Miltenyi Biotec, Germany). PBMCs were stimulated for 18 h at 37 °C in a 5% CO2 atmosphere in complete culture medium (RPMI 1640 supplemented with 10% fetal bovine serum and 1% each of L-glutamine, sodium pyruvate, nonessential amino acids, antibiotics, 0.1 M HEPES, 55 µM β-mercaptoethanol). For each stimulated sample, an unstimulated one was prepared, as negative control. After stimulation, cells were washed with PBS and stained with PromoFluor IR-840 (Promokine, PromoCell, Heidelberg, Germany) for 20 min at room temperature (RT). Next, cells were washed with FACS buffer (PBS supplemented with 2% FBS) and stained with the following fluorochrome-labeled mAbs: CXCR5-BUV661, CCR6-BUV496, CXCR3-BV785 for 30 min at 37 °C. Finally, cells were washed with FACS buffer and stained for 20 min at RT with Duraclone IM T cell panel (Beckman Coulter, Brea, CA) containing CD45-Krome Orange, CD3-APC-A750, CD4-APC, CD8-AF700, CD27-PC7, CD57-Pacific Blue, CD279 (PD1)-PC5.5, CD28-ECD, CCR7-PE, CD45RA-FITC and added with other three fluorescent mAbs i.e., CD69-BV650, CD137-BUV395 and CD95-BV605. Samples were acquired on a CytoFLEX LX flow cytometer (Beckman Coulter). All reagents used for T cell phenotyping are listed in Supplementary Table 1. All mAbs added to DuraClone IM T cells were previously titrated on human PBMCs and used at the concentration giving the best signal-to-noise ratio. The gating strategies used to identify CD4+ and CD8+ T cells are reported in the Supplementary Figs. 1 and 4.

### Detection of SARS-CoV-2-specific B cells
Thawed PBMC were washed twice with RPMI 1640 supplemented with 10% fetal bovine serum and 1% each of L-glutamine, sodium pyruvate, nonessential amino acids, anti- biotics, 0.1 M HEPES, 55 µM β-mercaptoethanol and 0.02 mg/ml DNASe. PBMC were washed with PBS and stained using viability marker PromoFluor IR-840 (Promokine, PromoCell, Heidelberg, Germany) for 20 min at RT in PBS. Next, cells were washed with PBS and stained for 15 min at RT with streptavidin-AF700 (decoy channel; ThermoFisher Scientific, USA) to remove false positive SARS-CoV-2-specific B cells. After washing with FACS buffer, cells were stained with biotinylated full-length SARS-CoV-2 spike protein (R&D Systems, Minneapolis) labeled with different streptavidin-fluorophore conjugates. Full-length biotinylated spike protein was mixed and incubated with streptavidin-BUV661(Becton Dickinson) or streptavidin-BV650 (BioLegend) at a 6:1 mass ratio for 15 min at RT. All samples were stained with both biotinylated streptavidin for 1 h at 4 °C. Then, cells were washed with FACS buffer and stained for 20 min at RT with DuraClone IM B cells (Beckman Coulter, Brea, CA) containing the following lyophilized directly conjugated mAbs: anti-IgD-FITC, CD21-PE, CD19-ECD, CD27-PC7, CD24-APC, CD38-AF750, anti-IgM-PB, CD45-KrO to which following drop-in antibodies were added: CD71-BUV395,

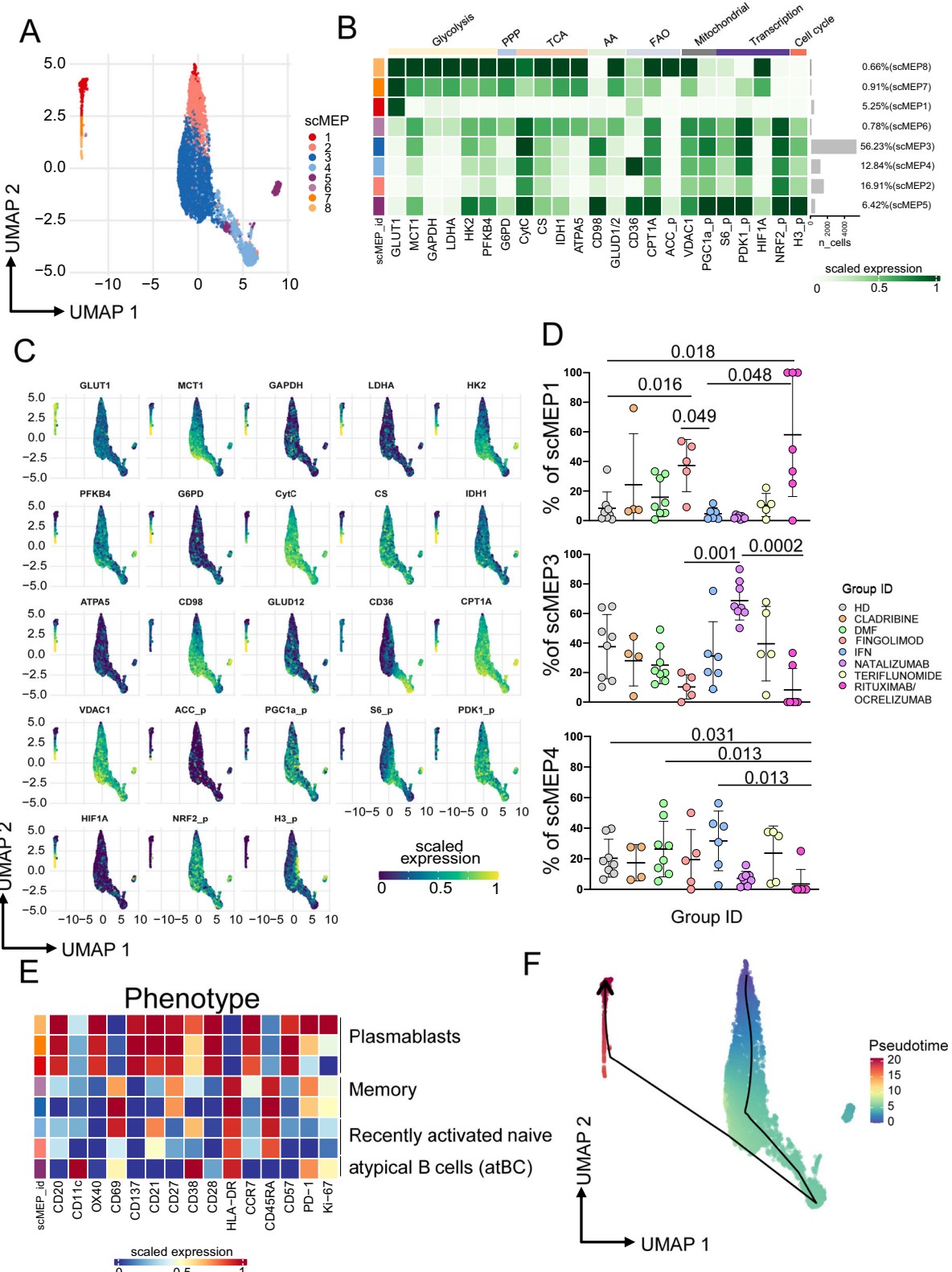

CD20-BV785, anti-IgG-BUV496 and anti-IgA-PerCP-Vio700. Samples were acquired on a CytoFLEX LX flow cytometer (Beckman Coulter). A minimum of 1,000,000 cells per sample were acquired. All reagents used for B cell phenotype are reported in Supplementary Table 3. All mAbs added to DuraClone IM B cells were previously titrated on human PBMCs and used at the concentration giving the best signal-to-noise ratio. The gating strategy used to identify Ag⁻ and Ag⁺ B cells is reported in the Supplementary Fig. 11.

### Intracellular cytokine staining (ICS)
Isolated PBMCs were thawed and rested for 6 h. PBMCs were stimulated in the presence of a pool of lyophilized peptides covering

**Fig. 6 | Ag$^+$ B cell metabolic states. A** Uniform Manifold Approximation and Projection (UMAP) plot shows the 2D spatial distribution of 9,780 cells from HD and MS patients. UMAP dimensionality reduction is calculated using sampled data from all cells and all available metabolic features; through FlowSOM clustering are identified 8 different clusters, defined as scMEP. Cells are colored by their scMEP state. **B** Heatmap of the median marker intensities of the 23 metabolic markers across the 8 cell populations, obtained with FlowSOM algorithm. The colors of cluster_id column correspond to the colors used to label the UMAP plot clusters/ scMEP. The color in the heatmap is referred to the median of the arcsinh marker expression (0–1 scaled): white represents a lower expression, while dark green represents a higher expression. Light gray bar along the rows (clusters/scMEP) and values in percentages indicate the relative sizes of the clusters. **C** Projection of UMAP graphs stratified by HD and all different MS patients. **D** Representative dot plots showing percentages of scMEP1, scMEP3 and scMEP4 among different

Group_IDs. The central bar represents the mean ± SD. Kruskal–Wallis test (one-sided) with Benjamini–Hochberg correction for multiple comparisons is used to test the differences among groups. Adjusted $q$-values are reported in the figure, if significant. healthy donors (HD, $n = 8$), multiple sclerosis patients treated with Cladribine ($n = 4$), Dimethyl Fumarate (DMF, $n = 8$), Fingolimod ($n = 5$), IFN ($n = 6$), Natalizumab ($n = 8$), Teriflunomide ($n = 5$), Rituximab/Ocrelizumab ($n = 7$). **E** Heatmap of 15 immunological markers enrichment modeling (not used for metabolic clustering) across different scMEP states, showing the relationship between metabolic states and functional properties. Light gray bar along the rows (clusters/scMEP) and values in percentages indicate the relative sizes of the clusters/scMEP. **F** Pseudotime visualization of scMEP development based on the estimated trajectory and envisaged in UMAP space. Colors are representative of the different distribution of cells population: blue represents less active metabolic state, while red represents an increased/enhanced metabolic state.

the complete protein coding sequence (aa 5–1273) of spike glycoprotein ("S") of SARS-CoV-2 (PepTivator SARS-CoV-2 Prot_S Complete Miltenyi Biotec, Bergisch Gladbach, Germany) together with 1 μg/ml of anti-CD28/49d (Becton Dickinson). PBMCs were stimulated for 16 h at 37 °C in a 5% $CO_2$ atmosphere in complete culture medium (RPMI 1640 supplemented with 10% FBS and 1% each of L-glutamine, sodium pyruvate, non-essential amino acids, antibiotics, 0.1 M HEPES, 55 mM β-mercaptoethanol, and 0.02 mg/mL DNAse I). For each stimulated sample, an unstimulated one was prepared as a negative control. All samples were incubated with protein transport inhibitors brefeldin A (Golgi Plug, Becton Dickinson Bioscience, San Jose, CA, USA) and monensin (Golgi Stop, Becton Dickinson Bioscience, San Jose, CA, USA) and previously titrated concentration of CD107a-PE (BioLegend, San Diego, CA, USA). After stimulation, cells were washed with PBS and stained with LIVE/DEAD fixable Aqua (ThermoFisher Scientific, USA) for 20 min at RT. Next, cells were washed with FACS buffer and stained with surface mAbs recognizing CD3-PE.Cy5, CD4-AF700, and CD8-APC.Cy7 (BioLegend, San Diego, CA, USA). Cells were washed with FACS buffer and fixed and permeabilized with the Cytofix/Cytoperm buffer set (Becton Dickinson Bioscience, San Jose, CA, USA) for cytokine detection. Then, cells were stained with previously titrated mAbs recognizing IL-17-PE-Cy7, TNF-BV605, IFN-γ-FITC, IL-2-APC, and GRZB-BV421 (all mAbs from BioLegend, San Diego, CA, USA). Samples were acquired on an Attune NxT acoustic cytometer (ThermoFisher Scientific, USA). Supplementary Table 2 reports mAb titers, clones, catalog numbers, and type of fluorochrome used in the panel.

Gating strategy used to identify and analyze the intracellular cytokine production of CD4$^+$ and CD8$^+$ T lymphocytes is reported in Supplementary Fig. 7.

### Computational analysis of flow cytometry data

**T cell analysis.** Compensated Flow Cytometry Standard (FCS) 3.0 files were imported into FlowJo software version v10.7.1 and analyzed by standard gating to remove doublets, aggregates and dead cells. For ex vivo immunophenotyping of non-antigen-specific (Ag$^−$) and antigen-specific (Ag$^+$) T cells of both CD4$^+$ and CD8$^+$ we analyzed only the data of stimulated samples. For each sample, we therefore selected data from all living CD4$^+$ or CD8$^+$ T cells and imported them in R using flowCore package v2.4.0 f or a total of 37,397,203 CD4$^+$ T cells (of which 465,729 were SARS-CoV-2 specific) and 12,758,008 CD8$^+$ T cells (of which 180,267 were SARS-CoV-2 specific). The further analysis was performed using CATALYST v1.17.3. All data obtained by flow cytometry were transformed in R using hyperbolic arcsine "arcsinh (x/cofactor)" applying manually defined cofactors (where x is the fluorescence measured intensity value). Clustering and dimensional reduction were performed using FlowSOM (version 2.4.0) and UMAP (version 0.2.8.0) algorithms, respectively. The Ag$^+$ CD4$^+$ and CD8$^+$ T cell clusters have been analyzed using the following markers: CD45RA, CCR7, CD27, CD28, PD-1, CCR6, CXCR3, CXCR5 and CD95. The quality

control (QC) of clustering for CD4$^+$ and CD8$^+$ T cells is reported in the respective Supplementary Figs. 2 and 5.

**B cell analysis.** Compensated Flow Cytometry Standard (FCS) 3.0 files were imported into FlowJo software version v10.7.1 and analysed by standard gating to remove doublets, aggregates, dead cells, and identify CD19$^+$ B cells. From total CD19$^+$ B cells, to remove false positive SARS-CoV-2-specific B cells we eliminated decoy-positive B cells. For each sample, we selected the SARS-CoV-2-specific B cells as positive cells for both Spike_streptavidin-BUV661 and Spike_streptavidin-BV650 and now referred to as Ag$^+$ B cells. The remaining double negative cells were non-SARS-CoV-2-specific B cells and mentioned to as Ag$^−$ B cells. Then, we exported for each sample separately both Ag$^+$ and Ag$^−$ B cells and imported them in R using flowCore package v2.4.0. The unsupervised analysis was performed using CATALYST v1.17.3. All data were transformed in R using hyperbolic arcsin (arcsinh x/cofactor) applying manually defined cofactors. Clustering and dimensional reduction were performed using FlowSOM and UMAP algorithms, respectively. For each day of acquisition at CytoFLEX LX, we had a sample used as quality control (QC).

### Mass cytometry

**scMEP staining protocol.** Thawed PBMCs were washed twice with RPMI 1640 supplemented with 10% fetal bovine serum and 1% each of L-glutamine, sodium pyruvate, nonessential amino acids, antibiotics, 0.1 M HEPES, 55 μM β-mercaptoethanol and 0.02 mg/ml DNAse. PBMCs were washed with Maxpar PBS and stained for 5 min at 37 °C with a working solution of the pre-titrated Cell-ID Cisplatin-195Pt in Maxpar PBS. For quenching the cisplatin stain, PBMCs were washed twice with Maxpar Cell Staining Buffer, using at least 5x the volume of the cell suspension. After that, PBMCs were stained with 100 μl of 1x Surface mAb Mix (see Supplementary Table 5 for all reagents used) at room temperature for 15 min. Samples were gently vortexed and incubated at room temperature for additional 15 min. Following the incubation, cells were washed twice by adding 2 mL Maxpar Cell Staining Buffer to each tube, centrifuged at $300 \times g$ for 5 min, and supernatant was removed by aspiration, leaving a residual volume about 100 μL. For each sample, the pellet was thoroughly disrupted by pulse vortex. Cells were prepared for nuclear staining, 1 mL of Foxp3 Fixation/Permeabilization working solution was added to each sample, and they were incubated for 30 min at 4 °C, protected by light. Then, 2 mL of 1X Permeabilization Buffer were added to each samples' tube and centrifuged at $400–600 \times g$ for 5 min at room temperature. PBMCs were stained with 100 μl of 1x Nuclear mAb Mix and incubate for at least 30 min at 4 °C. Cells were subsequently washed twice; a first time with 2 mL of 1X Permeabilization Buffer and a second time with 2 mL of Maxpar Cell Staining Buffer. Samples were placed on ice for 10 min to chill. Then, 1 mL of 4 °C methanol was added, samples were mixed gently, and incubated on ice for further 15 min. PBMCs were washed twice with 2 mL of Maxpar Cell Staining

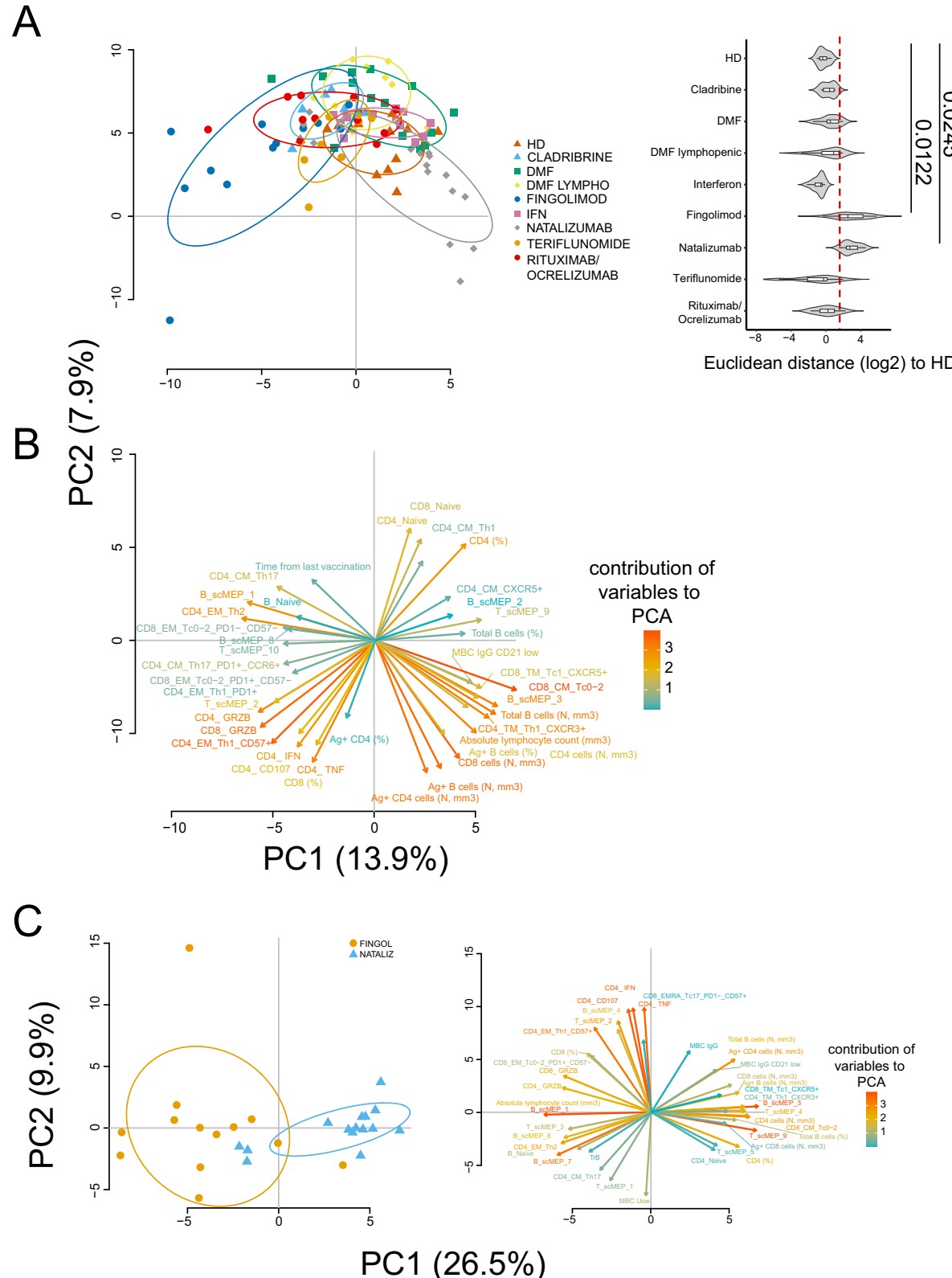

**Fig. 7 | Principal component analysis (PCA) of HD and MS treated groups. A** PCA showing the spatial distribution of vaccinated MS patients treated with different DMT and healthy donors (HD). Euclidean distance to HD has been calculated. Violin plot showing median, interquartile range (IQR) and whiskers (1.5*IQR). Kruskal−Wallis test (one-sided) with Benjamini−Hochberg correction for multiple comparisons is used to test the differences among groups, *p < 0.05. **B** Plot displaying the variables as vector, indicating the direction of each variable to overall distribution. The strength of each variable is represented by colors: orange color represents a strong contribution; light blue color represents a milder contribution. Length and direction of the arrows indicate the weight and correlation for each parameter. **C** (Left) PCA showing the spatial distribution of MS patients treated with fingolimod or natalizumab after SARS-CoV-2 vaccination, (Right) contribution of each immunological variables to PCA. Healthy donors (HD, *n* = 8), multiple sclerosis patients treated with Cladribine (*n* = 4), Dimethyl Fumarate (DMF, *n* = 8), Fingolimod (*n* = 5), IFN (*n* = 6), Natalizumab (*n* = 8), Teriflunomide (*n* = 5), rituximab/ocrelizumab (*n* = 7).

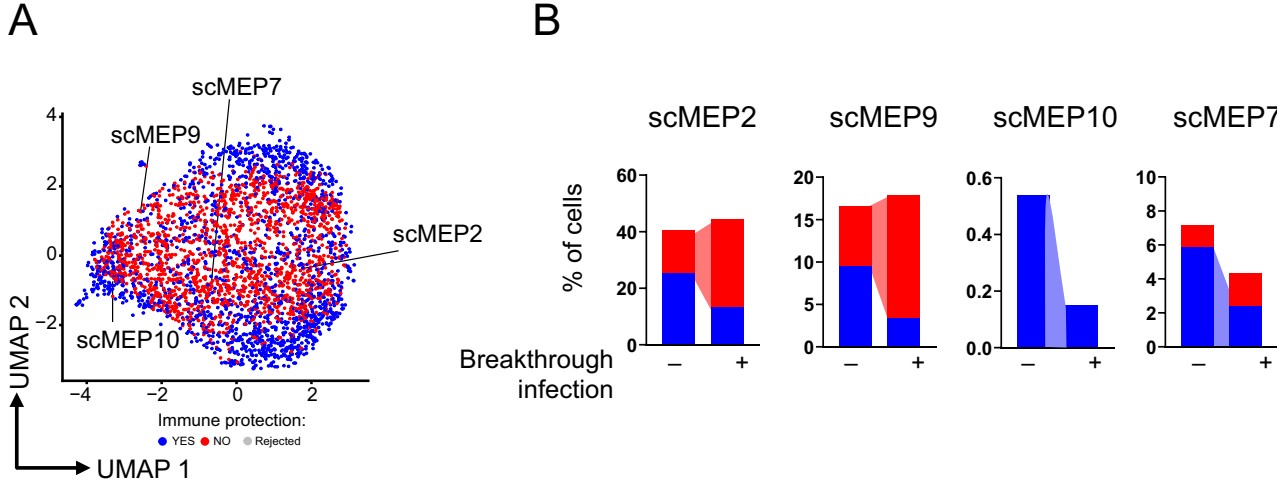

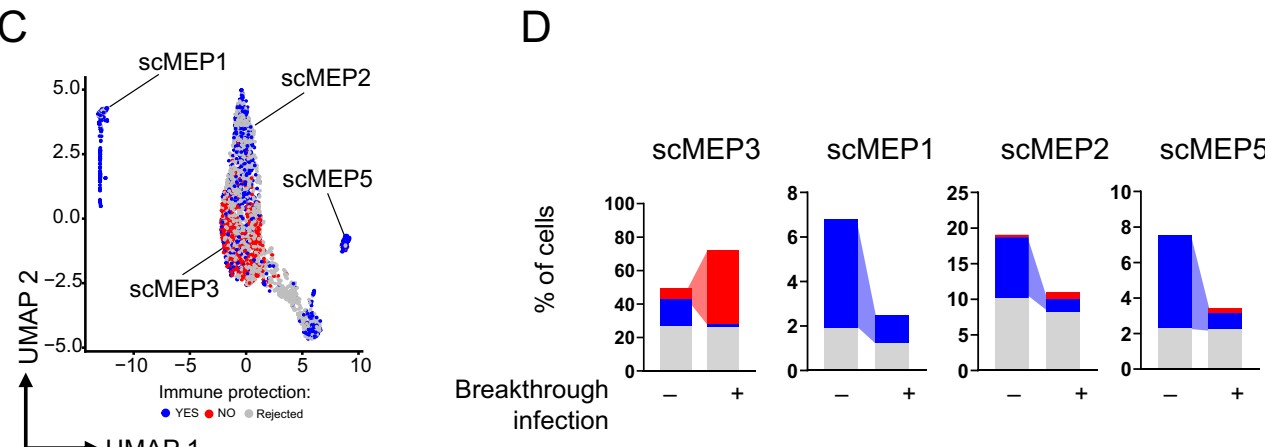

**Fig. 8 | PENCIL prediction of Ag⁺ T and B cell subpopulations associated with SARS-CoV-2 breakthrough infection. A** UMAP visualization displaying the Ag⁺ T cells from specific scMEP clusters. **B** Bar plot illustrating the percentage of cells within the scMEP clusters, whether associated or not with immune protection, among patients experiencing or not SARS-CoV-2 breakthrough infections. **C** UMAP visualization displaying the Ag⁺ B cells from specific scMEP clusters. **D** Bar plot illustrating the percentage of cells within the scMEP clusters, whether associated or not with immune protection, among patients experiencing or not SARS-CoV-2 breakthrough infections. In gray: not assigned cells (Rejected); in blue: cells associated with immune protection (YES); in red: cells not associated with immune protection (NO).

Buffer, centrifuged at $800 \times g$ for 5 min, and supernatants were removed by aspiration. PBMCs were then stained with 100 µl of 1x Phospho-Mix and incubated for 30 min at room temperature. Cells were washed twice by adding 2 mL Maxpar Cell Staining Buffer to each tube, centrifuged at $800 \times g$ for 5 min, and supernatants were removed by aspiration, leaving a residual volume about 100 µL. It was essential to thoroughly disrupt the pellet by pulse vortex, before adding 1 mL of the 1.6% formaldehyde solution to each tube. After gently mixing, PBMCs were incubated at room temperature for 10 min. When incubation was completed, cells were centrifuged at $800 \times g$ for 5 min and supernatant was removed by aspiration. One mL of Cell-ID Intercalation solution was then added to each sample and gently vortex. Samples were incubated at 4 °C overnight. After centrifuging tube at $800 \times g$ for 5 min, each pellet was resuspended in the residual volume of iridium fix/perm solution and transferred into a labeled 1.5 mL microcentrifuge tube ready to be stored at −80 °C.

After thawing in Maxpar cell staining buffer (CSB), cells were washed twice with CSB. Prior to acquisition cells were also washed with Maxpar cell acquisition solution (CAS). Immediately before acquisition, cells were resuspended to a final concentration of 10^6/ml in CAS with EQ Passport beads (1:10 dilution) and acquired on the Helios™ system (Standard Biotools, South San Francisco, CA, USA). Acquisition rate was constantly monitored at 350 to 400 events/sec, to minimize ion cloud fusion and maximize data quality. Acquired data were normalized using Passport beads (Fluidigm/Standard Biotools method) with CyTOF software (version 10.7.1014).

**SCENITH assay on SARS-CoV-2-specific B cells.** Freshly isolated PBMCs were rested overnight in RPMI 1640 supplemented with 10% FBS and 1% each of L-glutamine, sodium pyruvate, nonessential amino acids, antibiotics, 0.1 M HEPES, 55 mM β-mercaptoethanol. After resting, cells were washed with PBS and stained with biotinylated full-length SARS-CoV-2 spike protein (R&D Systems, Minneapolis, USA)

labeled with different streptavidin fluorophore conjugates. Full-length biotinylated spike protein was mixed and incubated with streptavidin-BUV661 (Becton Dickinson) or streptavidin-BV650 (BioLegend) at a 6:1 mass ratio for 15 min at room temperature (RT). PBMCs were stained with both biotinylated streptavidin at 4 °C for 1 h. Then, cells were washed at 1000 rpm for 7 min and resuspended in complete medium at a density of $1 \times 10^6/0.1$ ml and incubated at 37 °C for 4 h. After the incubation cells were washed at 1000 rpm for 7 min and resuspended in 340 µl of complete medium and equally distributed in 4 tubes (one for each condition CO, DG, DGO, O) to proceed with SCENITH protocol. All the reagents were prepared the day of the experiment, empty tubes and 20X inhibitors CO; DG; O; P (stored at −20 °C in aliquots) were equilibrated at 37 °C for 30 min before cell treatment. 5 µl of each 20X inhibitor and 10 µl of 20X puromycin were added to the corresponding tube and incubated at 37 °C for 40 min. In the DGO tube, DG and O were added simultaneously. After incubation, tubes with cells were filled up with ice cold MACS Buffer, centrifuged at 400 x g during 5 min at 4 °C and the supernatant was discarded by aspiring. Cells were resuspended in 100 µl of PromoFluor IR-840 (Promokine, PromoCell, Heidelberg, Germany) and Fc Block (Becton Dickinson) and incubated for 15 min, at 4 °C, in the dark. Without washing, 100 µl of 2X surface staining mix including the previously titrated mAbs CD19-PE, CD69-FITC, were added to cells and samples were incubated for 25 min at 4 °C in the dark. Tubes were filled up with FACS buffer, centrifuged at $400 \times g$, during 5 min, at 4 °C and the supernatant was discarded by aspiring. Red blood cells lysis was avoided. The following intracellular staining with Invitrogen FOXP3 stain buffer was performed: cells were resuspended in 100 µL of Foxp3 Fixation/Permeabilization solution, vortexed and incubated for 20 min at RT in the dark. Then 1X Permeabilization Buffer was added to cells and samples were centrifuged at $600 \times g$ for 5 min at RT. The supernatant was discarded by aspiration and cells were resuspended in 50 µL of intracellular block (1X Permeabilization Buffer + 20% FCS) and incubate for 10 min at RT. Without washing 50 µL of anti-puromycin-AF647 antibody solution 1/250 were added to cells and cells were incubated for 1 h at 4 °C in the dark. At the end of incubation cells were washed with 1X Permeabilization Buffer and centrifuged at $600 \times g$ for 5 min at 4 °C. The supernatant was discarded, and stained cells were resuspended in 400 µL of FACS Buffer and acquired by flow cytometer CytoFLEX LX (Beckman Coulter, Hialeah, FL). A minimum of 1,500,000 cells per sample were acquired. All reagents used for the staining of cells are reported in Supplementary Table 4.

**SCENITH assay performed on SARS-CoV-2-specific T cells.** Freshly isolated PBMCs were incubated at a density of $1 \times 10^6/0.1$ ml in complete medium (RPMI 1640 supplemented with 10% FBS and 1% each of L-glutamine, sodium pyruvate, nonessential amino acids, antibiotics, 0.1 M HEPES, 55 mM ß-mercaptoethanol) with CD40-blocking antibody (0.5 µg/ml final concentration) (Miltenyi Biotec, Bergisch Gladbach, Germany) for 15 min at 37 °C before stimulation. Then, cells were stimulated by adding in the medium PepTivator SARS-CoV-2 Prot_S complete (Miltenyi Biotec, Bergisch Gladbach, Germany) containing 15-mer peptides with 11-amino acid overlap, covering the complete sequence of Wuhan SARS-CoV-2 spike glycoprotein, together with CD28/CD49d (Becton Dickinson) and incubated for 18 h at 37 °C in a 5% CO2 atmosphere. At the end of stimulation, cells were washed at 1000 rpm for 7 min and resuspended in 340 µl of complete medium and equally divided into 4 FACS tubes (one for each condition CO, DG, DGO, O) to perform the SCENITH protocol. All the reagents were prepared the day of the experiment, empty tubes and 20X inhibitors CO; DG; O; P (stored at −20 °C in aliquots) were equilibrated at 37 °C for 30 min before cell treatment. 5 µl of each 20X inhibitor and 10 µl of 20X puromycin were added to the corresponding tube and incubated at 37 °C for 40 min. In the DGO tube, DG and O were added simultaneously. After incubation, tubes with cells were filled up with ice cold

MACS Buffer, centrifuged at $400 \times g$ during 5 min at 4 °C and the supernatant was aspirated. Cells were resuspended in 100 µl of PromoFluor IR-840 (Promokine, PromoCell, Heidelberg, Germany) and Fc Block (Becton Dickinson) and incubated for 15 min at 4 °C, in the dark. Without washing, 100 µl of 2X surface staining mix including the previously titrated mAbs CD4-FITC, CD8-PE, CD3-PB, CD69-BV650, CD137-BUV395 were added to cells and samples were incubated for 25 min at 4 °C in the dark. Tubes were filled up with FACS buffer, centrifuged at $400 \times g$, during 5 min, at 4 °C and the supernatant was discarded by aspiring. Red blood cells lysis was avoided, and the following intracellular staining with Invitrogen FOXP3 stain buffer was performed: cells were resuspended in 100 µL of Foxp3 Fixation/Permeabilization solution, vortexed and incubated for 20 min at RT in the dark. Then 1X Permeabilization Buffer was added to cells and samples were centrifuged at $600 \times g$ for 5 min at RT. The supernatant was discarded by aspiring and cells were resuspended in 50 µL of intracellular block (1X Permeabilization Buffer + 20% FCS) and incubated for 10 min at RT. Without any washing step, 50 µL of anti-puromycin-AF647 antibody solution 1/250 was added to cells and followed by incubation for 1 h at 4 °C in the dark. Cells were washed with 1X Permeabilization Buffer and centrifuged at 600 g for 5 min at 4 °C. The supernatant was discarded, and stained cells were resuspended in 400 µl of FACS Buffer and acquired by flow cytometer CytoFLEX LX (Beckman Coulter, Hialeah, FL). A minimum of 1,500,000 cells per sample were acquired. All reagents used for the staining of cells are reported in Supplementary Table 4.

**Statistical analysis.** Quantitative variables were compared using Kruskal-Wallis non-parametric test corrected for multiple comparisons by controlling the False Discovery Rate (FDR), method of Benjamini and Hochberg. Statistically significant q-values are represented. Statistical analysis of cytokines production was performed using GraphPad Prism version 8 (GraphPad Software Inc., La Jolla, USA). Total percentage of antigen-specific ($Ag^+CD4^+$ and $Ag^+CD8^+$) T cell data have been calculated as background subtracted data. Simplified Presentation of Incredibly Complex Evaluation (SPICE) software (version 6, Vaccine Research Center, NIAID, NIH, Bethesda, MD, USA) was used to analyze flow cytometry data on T cell polyfunctionality. Data from the total cytokine production are represented as individual values, means, and standard errors of the mean. Regarding polyfunctionality, data in pie charts are represented as median values and statistical analysis was performed using permutation test; data in graphs are represented as individual values, means, and standard errors of the mean.

**Principal component analysis.** Principal Component Analysis (PCA) was executed and visualized in R using the prcomp function (stats v3.6.2) and the pca3d package v0.1. The data used included the proportions, absolute number and scMEP matabolic state of $Ag^+$ $CD19^+$ B cells, $CD4^+$, $CD8^+$ T cells along with clinical parameters (reported in Source Data File). Missing values of dataset were imputed using missMDA package v1.18. The total impact of a specific variable retained by PC1 and PC2 was computed as $[(C1 * Eig1) + (C2 * Eig2)]/(Eig1 + Eig2)$, where C1 and C2 represent the contributions of the variable to PC1 and PC2, and Eig1 and Eig2 denote the eigenvalues of PC1 and PC2, respectively. The Euclidean distance of MS-treated groups to HD in PCA space was calculated using the phenoptr v.0.3.2 package.

**PENCIL prediction analysis.** PENCIL v0.7 was used to predict cell clusters associated with absence of breakthrough infection in MS patients and HD[41]. As single cell input data we used our 45-parameter mass cytometry data (scMEP data) analyzed previously with R by using CATALYST v1.18.1 (see method above). We imported into Seurat v4.9.9 58 the expression matrix, containing hyperbolic arcsinh (cofactor 5) transformed data, and the metadata (also UMAP coordinates) of Ag+ T

or B lymphocytes. Cells from all individuals were divided in two groups on the basis of SARS-CoV-2 breakthrough infection and were used as input to run PENCIL. For T cells prediction, to reduce the number of rejected cells due to huge differences in cell number form who experienced SARS-CoV-2 breakthrough infection (10,591 cells) and who did not (96,931 cells), a downsampled dataset was used (a total of 21,591, whose 11,000 for who experienced SARS-CoV-2 breakthrough infection; 10,591 for who did not get COVID-19). Prediction displayed 68% accuracy. Then 5650 was used for Ag$^+$ B cells prediction (2825 for each group); a total of 1047 cells were rejected. The precision of the prediction was 76%.

## Reporting summary

Further information on research design is available in the Nature Portfolio Reporting Summary linked to this article.

## Data availability

All data generated or analysed in this study are included in this published article (and its supplementary information files). Further inquiries can be directed to the corresponding authors. Source data are provided with this paper.

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

## Acknowledgements

This work was supported by grants from: Fondazione Italiana Sclerosi Multipla (FISM) to A.C., project "Unraveling the role and functionality of T cells in treated multiple sclerosis SARS-CoV-2 vaccinated patients," protocol no. 2021/C19-R-Single/011; Bando MIUR "Dipartimenti di Eccellenza 2023/2027", Area CUN_06 Scienze Mediche, to A.C.; S.D.B. and L.Gi are Marylou Ingram Scholar of the International Society for Advancement of Cytometry (ISAC) for the period 2015–2020 and 2020–2025, respectively. Drs. Paola Paglia (ThermoFisher Scientific, Monza, Italy), Leonardo Beretta (Beckman Coulter, Milan, Italy), Dr. Paolo Santino, Ernesto Lopez, Gloria Martrus (Standard Biotools, San Francisco, CA, US), Dr. Marco Mattioli, Dr. Andrea De Fanti and Dr. Alessia Di Nella are acknowledged for their support in providing reagents and materials, for precious help and technical suggestions. Finally, we gratefully acknowledge the individuals who donated blood to participate in this study.

## Author contributions

A.N., R.B., N.P., A.P., E.S., A.L.C. and T.T. performed experiments; M.Cu, T.T. quantified plasmatic antibodies; D.L.T., M.R. performed bioinformatic analysis; D.F., F.V., M.Ca enrolled the patients; L.G., R.J.A., I.R., A.C. discussed the data; S.D.B. and A.C. supervised experiments, bioinformatic analysis, and wrote the manuscript; A.C. and D.F. corrected and revised the manuscript.

## Competing interests

There are restrictions to the commercial use of SCENITH due to a pending patent application by R.J.A. (PCT/EP2020/060486). A.N., R.B., N.P., A.P., E.S., M.Cu, A.L.C., M.C., T.T., D.L.T., M.R., D.F., F.V., M.Ca, L.G., I.R., A.C., S.D.B., D.F. declare that the research was conducted in the absence of any commercial or financial relationships that could be construed as a potential conflict of interest.
