## [Peer Review File · Nature Communications]

Immunosenescence and vaccine efficacy revealed by immunometabolism of SARS-CoV-2-specific cells in vaccinated Multiple Sclerosis patientsREVIEWER COMMENTS

Reviewer #1 (Remarks to the Author):

this works combines exhaustive immune and metabolomic phenotyping to explore immune responses of MS patients treated with different disease modifying drugs and healthy controls to third dose of anti SARS-CoV2 vaccine. Additionally, the use of a bioinformatic tool as PENCIL allowed to predict the protective effect of the vaccine in the different groups of patients. The combination of immune and metabolomic phenotypes is highly interesting since give a more complete idea of the activation status, and the use of PENCIL extend the use of this tool to this kind of studies.

However, showing the multiple analyses of the immune and metabolomic responses of patients treated with different drugs exhaustively, makes difficult to find the important results. I think it is important to compare every response to healthy individuals and then to deepen in those given different results i.e. Natalizumab and fingolimod.

For sake of clarity, I suggest shortening results and supplementary material.

Reviewer #2 (Remarks to the Author):

The study by Biasi et al aimed to assess the phenotype and function of SARS-CoV-2-specific T and B cells after mRNA vaccination in a cohort of MS patients. The intention was to study how disease modifying therapies (DMT) impact mRNA vaccination and analyse if any peculiar immune response could be further used to predict protection from breakthrough infections. The study was well conducted and detail but remain observational. Some specific comments:

1. Did the authors use another vaccinated cohort? These could include non-mRNA COVID-19 vaccines, or vaccines against other diseases such as flu vaccines?
2. The cohort is skewed towards females (~72%), and would likely have an impact on the distribution. Can the authors please provide more insights if recruiting more males is not possible?
3. Did the authors confirm that during the collection after 6 to 8 months of vaccination, the patients did not suffer from other infections or ailments? This is an important factor to note as it would affect the immune profiles of the B and T cells.

Reviewer #3 (Remarks to the Author):

De Biasi et al. examine the effects of immunomodulatory therapy in MS patients on their COVID-19 vaccine responses. They find detectable and functional responses in almost all patients. However, they identify metabolic profiles that differ between treatment groups. Furthermore, they describe a specific signature that may predict protection from breakthrough infection.

While the total number of patients studied is reasonable, individual treatment groups each contain only a handful of patients; and there is considerable variability, for example, among absolute CD4 counts in natalizumab-treated individuals (Figure 1A). It's not clear why some differences are seen via absolute counts but not percentages, or vice versa; but this intra-group variability may be play into it. It probably also plays a role in the many "significant" findings that don't look to be dramatically different. In fact, the inter-group differences in phenotype are in general not well explained, just observations that may or may not hold up in a larger group analysis (or with stricter control of multiple comparisons). An exception is the clear lack of antigen-specific B cells with anti-CD20 treatment, which is expected.

An interesting aspect of their analysis is the use of metabolic profiling by CyTOF. They find differing metabolic states between treatment groups, and relate these to phenotypic profiles. When comparing overall signatures, two groups, Fingolimod- and Natalizumab-treated, appear different from healthy donors. There is at least some discussion on the likely mechanisms for these

differences.

Finally, the authors show some correlations of specific metabolic signatures with presence or absence of breakthrough infection. This is speculative at best, given the fact that there were only 18 breakthrough infections, but that lack thereof does not guarantee a lack of susceptibility (since exposure was not controlled). I also don't understand the red and blue designations in these "PENCIL" analyses. It seems that some cells are designated as protective and some not, but they are all intermingled in the UMAP, and they don't segregate by metabolic profile (except scMEP10, which appears to be entirely protective).

RESPONSE TO REVIEWERS' COMMENTS

Reviewer #1 (Remarks to the Author):

This work combines exhaustive immune and metabolomic phenotyping to explore immune responses of MS patients treated with different disease modifying drugs and healthy controls to third dose of anti SARS-CoV2 vaccine. Additionally, the use of a bioinformatic tool as PENCIL allowed to predict the protective effect of the vaccine in the different groups of patients. The combination of immune and metabolomic phenotypes is highly interesting since give a more complete idea of the activation status, and the use of PENCIL extend the use of this tool to this kind of studies.

However, showing the multiple analyses of the immune and metabolomic responses of patients treated with different drugs exhaustively, makes difficult to find the important results. I think it is important to compare every response to healthy individuals and then to deepen in those given different results i.e. Natalizumab and fingolimod.

For sake of clarity, I suggest shortening results and supplementary material.

We thank the reviewer for the comments. We amended as requested and the manuscript has been modified accordingly.

Reviewer #2 (Remarks to the Author):

The study by De Biasi et al aimed to assess the phenotype and function of SARS-CoV-2-specific T and B cells after mRNA vaccination in a cohort of MS patients. The intention was to study how disease modifying therapies (DMT) impact mRNA vaccination and analyse if any peculiar immune response could be further used to predict protection from breakthrough infections. The study was well conducted and detail but remain observational. Some specific comments:

1. Did the authors use another vaccinated cohort? These could include non-mRNA COVID-19 vaccines, or vaccines against other diseases such as flu vaccines?

We thank the reviewer for the comment. We did not take other vaccinated cohorts into account. Regarding the COVID-19 vaccination booster dose, as per recommendations of the Italian Society of Neurology in conjunction with the Italian MS Association (<https://www.aism.it/raccomandazioni-covid-19-persone-con-sclerosi-multipla-aism-sin#vaccinazioni>), only mRNA vaccines were administered to MS patients. However, in a previous study by our group, we did compare immunological response developed by healthy donors receiving different combination of COVID-19 vaccinations (mRNA vaccination and heterologous vaccination, such as DNA vaccine followed by mRNA vaccination). We deeply characterized the SARS-CoV-2 B and T cell repertoire and we did not find any differences in terms of phenotype and function between these two groups (Lo Tartaro D., Detailed characterization of SARS-CoV-2-specific T and B cells after infection or heterologous vaccination. *Front Immunol.* 2023 Feb 9;14:1123724. doi: 10.3389/fimmu.2023.1123724.)

Regarding flu vaccination, we did not take MS patients receiving flu vaccines, nor the antigen-specific response, into consideration since our focus was on COVID-19 due to its deep impact on morbidity and mortality in specific MS cohorts. Furthermore, the proportion of patients who carried out the flu vaccine throughout the enrolment period did so on the same day of the COVID-19 vaccine (which was carried out by virtually all patients), so we would not have been able to enroll a cohort who uniquely received the flu vaccine. Regarding this topic, MS patients receiving immunomodulatory treatment had reduced protection (27.4%), compared to controls (43.5%), after pandemic H1N1 vaccination (2009). The rates of protection were not influenced by interferon beta treatment (44.4% protected), but were reduced among patients receiving glatiramer acetate (21.6%), natalizumab (23.5%), and mitoxantrone (0.0%). A similar pattern emerged after MS patients received a seasonal influenza vaccination in 2010 (DOI: 10.1177/1352458513513970). However, a systematic review and meta-analysis investigating the immunogenicity of the influenza vaccine in MS patients revealed that there was no statistical difference in immune response mounted against the influenza vaccine between

healthy controls and multiple sclerosis patients (Nguyen J, Immunogenicity of The Influenza Vaccine in Multiple Sclerosis Patients: A Systematic Review and Meta-Analysis. *Mult Scler Relat Disord*. 2021 Feb;48:102698. doi: 10.1016/j.msard.2020.102698).

2. The cohort is skewed towards females (~72%), and would likely have an impact on the distribution. Can the authors please provide more insights if recruiting more males is not possible?

We thank the reviewer for the point raised, but the study cohort precisely reflects the actual ratio of autoimmune diseases. Indeed, in multiple sclerosis and myasthenia gravis, the female to male ratio is 2:1-3:1. It is also true that symptom severity, disease course, response to therapy and overall survival may differ between males and females with autoimmune diseases. Overall, women mount stronger humoral and cellular immune responses than men and this is believed to impact on the different susceptibility to develop autoimmune diseases. The main factors affecting the differences between female and male immune systems are the sex hormones and the different response to environmental factors, such as microbial exposure and diet. Indeed, in PBMCs from patients with multiple sclerosis, females show greater activation of TH1 cells and increased levels of IFN γ production, whereas males exhibit greater TH17 cell responses owing to androgen receptor regulation of PPAR α expression in T cells (Zhang, M. A. et al. Peroxisome proliferator-activated receptor (PPAR) α and γ regulate IFN γ and IL-17A production by human T cells in a sex-specific way. *Proc. Natl Acad. Sci. USA* 109, 9505–9510 (2012).

Unluckily, it was not possible to recruit additional males with multiple sclerosis to be enrolled in the study for two reasons. The first is that ethical committee approval is ended; the second is that it would be difficult to match the time (months) after third dose vaccination and exclude patients who experienced breakthrough infections or disease relapses. It is true that the cohort of patients is small, but it is precisely matched and balanced.

3. Did the authors confirm that during the collection after 6 to 8 months of vaccination, the patients did not suffer from other infections or ailments? This is an important factor to note as it would affect the immune profiles of the B and T cells.

We thank the reviewer for the comment. Patients are seen at least every six months, when information on any intervening pathologies/medical issues/concomitant drugs are collected. All patients' charts were newly reviewed and did not yield any information on other infections or ailments.

Reviewer #3 (Remarks to the Author):

De Biasi et al. examine the effects of immunomodulatory therapy in MS patients on their COVID-19 vaccine responses. They find detectable and functional responses in almost all patients. However, they identify metabolic profiles that differ between treatment groups. Furthermore, they describe a specific signature that may predict protection from breakthrough infection.

While the total number of patients studied is reasonable, individual treatment groups each contain only a handful of patients; and there is considerable variability, for example, among absolute CD4 counts in natalizumab-treated individuals (Figure 1A). It's not clear why some differences are seen via absolute counts but not percentages, or vice versa; but this intra-group variability may be play into it. It probably also plays a role in the many "significant" findings that don't look to be dramatically different. In fact, the inter-group differences in phenotype are in general not well explained, just observations that may or may not hold up in a larger group analysis (or with stricter control of multiple comparisons). An exception is the clear lack of antigen-specific B cells with anti-CD20 treatment, which is expected.

With regard to the cohort size, and in particular to the low numerosity of some of the treatment groups, we underline the following points. Throughout the COVID-19 pandemic, blood samples following SARS-

CoV-2 vaccination were collected from a total of 289 Multiple Sclerosis patients: 14 on cladribine, 17 on teriflunomide, 23 on beta-interferon 1a, 37 on fingolimod, 48 on anti-CD20 agents, 71 on natalizumab and 79 on dimethylfumarate. The different drug frequencies mirror clinical practice drug prescriptions of that time-period which are based on a number of variables including disease activity, contraindications, convenience, safety profile, tolerability and also regulatory agency (Agenzia Italiana del Farmaco, AIFA) limitations. Patients included in this study were accurately chosen in order to have homogeneous clinical phenotypes (exclusion on primary/secondary progressive MS, in particular for those treated with anti-CD20), the same number of SARS-CoV-2 vaccine doses and, not less importantly, a similar timing between third vaccine dose and sampling. As a consequence, the final number of enrolled patients was relatively low for some of the less prescribed drugs.

Regarding the **intra variability**, we would like to underline that there were no outliers within any group. Before performing statistical analysis, we checked the presence of outliers by using the “robust outlier removal” (ROUT) method for identifying these cases. This method combines “Robust regression” and “Outlier removal” under Prism 8.0. The ROUT method of regression follows three steps. First, robust nonlinear regression method is used to fit a curve that is not influenced by outliers. Second, the residuals of the robust fit are analyzed to identify any outliers. This step uses a new outlier test adapted from the False Discovery Rate approach of testing for multiple comparisons. Third, it removes the outliers, and perform ordinary least-squares regression on the remaining data. Here below, we report all the ROUT results for the main populations of interest that support our main conclusion. Where outlier values were present, all the statistical analyses have been performed on cleaned data.

% of Ag+ B cells

	HC	CLADRIBINE E	DMF	DMF LYMPHO	FINGOLIMO D	IFN	NATALIZUMAB	TERFLUONOMID E	anti-CD20
Method									
ROUT (Q = 1%)									
Number of points									
# Y values analyzed	12	5	12	9	12	13	15	7	9
Outliers	0	0	0	0	0	1	0	0	1

% of Ag+ CD4+ T cells

	HC	CLADRIBINE E	DMF	DMF LYMPHO	FINGOLIMO D	IFN	NATALIZUMAB	TERFLUNOMIDE	anti-CD20
Method									
ROUT (Q = 1%)									
Number of points									
# Y values analyzed	13	6	14	10	13	12	14	8	11
Outliers	0	0	2	0	0	0	0	0	2

Absolute number of Ag+ CD4+ T cells

	HC	CLADRIBIN E	DMF	DMF LYMPHO	FINGOLIMO D	IFN	NATALIZUMAB	TERFLUONOMID E	ANTI- CD20
Method									
ROUT (Q = 1%)									
Number of points									
# Y values analyzed	13	5	13	10	13	9	5	7	10
Outliers	0	0	0	0	1	0	0	0	0

% of Ag+ CD8+ T cells

	HC	CLADRIBIN E	DMF	DMF LYMPHO	FINGOLIMO D	IFN	NATALIZUMAB	TERFLUONOMID E	anti- CD20
Method									
ROUT (Q = 1%)									
Number of points									
# Y values analyzed	13	6	14	9	13	13	15	8	11
Outliers	0	0	1	0	1	0	0	1	0

scMEP 1 B cells

	HC	CLADRIBINE	DMF	FINGOLIMOD	IFN	NATALIZUMAB	TERFLUONOMIDE	ANTI-CD20
Method								
ROUT (Q = 1%)								
Number of points								
# Y values analyzed	8	4	8	5	6	8	5	7
Outliers	1	1	0	0	0	0	0	0

scMEP 3 B cells

	HC	CLADRIBINE	DMF	FINGOLIMOD	IFN	NATALIZUMAB	TERFLUONOMIDE	ANTI-CD20
Method								
ROUT (Q = 1%)								
Number of points								
# Y values analyzed	8	4	8	5	6	8	5	7
Outliers	0	0	0	0	0	0	0	2

scMEP 4 B cells

	HC	CLADRIBINE	DMF	FINGOLIMOD	IFN	NATALIZUMAB	TERFLUONOMIDE	ANTI-CD20
Method								
ROUT (Q = 1%)								
Number of points								
# Y values analyzed	8	4	8	5	6	8	5	7
Outliers	0	0	0	0	0	0	0	1

scMEP 2 B cells

	HC	CLADRIBINE	DMF	FINGOLIMOD	IFN	NATALIZUMAB	TERFLUONOMIDE	ANTI-CD20
Method								
ROUT (Q = 1%)								
Number of points								
# Y values analyzed	8	4	8	5	6	8	5	7
Outliers	0	0	0	0	0	0	0	0

scMEP 5 B cells

	HC	CLADRIBINE	DMF	FINGOLIMOD	IFN	NATALIZUMAB	TERFLUONOMIDE	ANTI-CD20
Method								
ROUT (Q = 1%)								
Number of points								
# Y values analyzed	8	4	8	5	6	8	5	7
Outliers	1	0	1	0	0	2	0	1

scMEP 2 T cells

	HC	CLADRIBINE	DMF	FINGOLIMOD	IFN	NATALIZUMAB	TERFLUONOMIDE	ANTI-CD20
Method								
ROUT (Q = 1%)								
Number of points								
# Y values analyzed	8	4	8	5	6	8	5	7
Outliers	0	0	0	0	0	0	0	0

scMEP 4 T cells

	HC	CLADRIBINE	DMF	FINGOLIMOD	IFN	NATALIZUMAB	TERFLUONOMIDE	ANTI-CD20
Method								
ROUT (Q = 1%)								
Number of points								
# Y values analyzed	8	4	8	5	6	8	5	7
Outliers	0	0	0	0	1	0	0	0

scMEP 9 T cells

	HC	CLADRIBINE	DMF	FINGOLIMOD	IFN	NATALIZUMAB	TERFLUONOMIDE	ANTI-CD20
Method								
ROUT (Q = 1%)								
Number of points								
# Y values analyzed	9	5	9	6	7	9	6	8
Outliers	0	0	0	0	0	0	0	1

scMEP 10 T cells

	HC	CLADRIBINE	DMF	FINGOLIMOD	IFN	NATALIZUMAB	TERFLUONOMIDE	ANTI-CD20
Method								
ROUT (Q = 1%)								
Number of points								
# Y values analyzed	8	4	8	5	6	8	5	7
Outliers	0	0	0	0	0	0	0	0

scMEP 7 T cells

	HC	CLADRIBINE	DMF	FINGOLIMOD	IFN	NATALIZUMAB	TERFLUONOMIDE	ANTI-CD20
Method								
ROUT (Q = 1%)								
Number of points								
# Y values analyzed	8	4	8	5	6	8	5	7
Outliers	0	0	1	0	0	0	0	0

Regarding **inter variability**, as indicated in the Method section, quantitative variables (namely, the percentages and absolute numbers of all the T and B cell populations taken into account) were compared using Kruskal-Wallis non-parametric test corrected for multiple comparisons by controlling the False Discovery Rate (FDR), according to the method of Benjamini and Hochberg. The Benjamini-Hochberg procedure, also known as the False Discovery Rate (FDR) procedure, is a statistical method used in multiple hypothesis testing to control the expected proportion of false discoveries. The Benjamini-Hochberg procedure addresses this issue by controlling the FDR, which is defined as the expected proportion of false positives among the rejected hypotheses. The calculation of adjusted p-values in the Benjamini-Hochberg procedure involves comparing each individual p-value to a critical value or threshold. The critical value is determined based on the desired false discovery rate (FDR) control, which in our case was 0.05. Statistically significant q-values are represented in the figures.

To conclude, unfortunately we cannot add more samples to our study, because at present it is almost impossible to match the time (months) after the third dose of vaccination, and to exclude patients who experienced breakthrough infections or disease relapses. We agree that the cohort of patients is relatively small, but it has been precisely matched and balanced. Moreover, even if in the results and conclusion we do not dedicate space enough to MS patients treated with cladribine and teriflunomide, we preferred to keep these data for the completeness of the manuscript.

An interesting aspect of their analysis is the use of metabolic profiling by CyTOF. They find differing metabolic states between treatment groups, and relate these to phenotypic profiles. When comparing overall signatures, two groups, Fingolimod- and Natalizumab-treated, appear different from healthy donors. There is at least some discussion on the likely mechanisms for these differences.

Finally, the authors show some correlations of specific metabolic signatures with presence or absence of breakthrough infection. This is speculative at best, given the fact that there were only 18 breakthrough infections, but that lack thereof does not guarantee a lack of susceptibility (since exposure was not controlled). I also don't understand the red and blue designations in these "PENCIL" analyses. It seems that some cells are designated as protective and some not, but they are all intermingled in the UMAP, and they don't segregate by metabolic profile (except scMEP10, which appears to be entirely protective).

We thank the reviewer for this comment. We agree that 18 breakthrough infections on a cohort of 101 subjects is not a big number, but PENCIL is a solid method of prediction. Indeed, in the original paper that describes PENCIL this prediction method has been validated on a cohort of patients with melanoma consisting of 17 responders and 31 non-responders to immune checkpoint blockade therapy. In this case, the prediction could be considered as a proof-of-concept application and meaning. Indeed, it is important to remark that for the first time PENCIL was applied to flow cytometry data, and in particular that it was used for studying COVID-19 vaccination in multiple sclerosis patients. Thus, we wanted to show that a likely metabolically active and experienced Ag-specific immune response correlates with the absence of breakthrough infection, and may thus confer high immune protection. Of course, this method could pave the way for prediction studies in larger cohort of patients and even in other vaccination campaigns, such as seasonal flu.

Regarding a possible exposure to SARS-CoV-2 of our patients, we agree with the reviewer, but it was likely impossible to control neither the exposure nor the infection because during the timing of patients' enrollment (from September 2021) there was no lockdown, and no mandatory COVID-19 testing.

The designation in PENCIL analysis is as follow:

- 1) cells associated with immune protection in blue, therefore enriched in subjects who did not develop breakthrough infection (YES in the figure legend number 8);
- 2) cells not associated with immune protection in red, therefore enriched in subjects who develop breakthrough infection (NO in the figure legend number 8);
- 3) cells displaying uncertain phenotype that cannot be assigned to one of the two categories (in our case immune protection YES or NO) were rejected to reduce the classification errors and are in grey. Red and blue cells are mixed in the UMAP graph due to the fact that prediction accuracy is not 100%, but rather approximately 80%. This implies that 20% cells from subjects who do not experience a breakthrough infection were not accurately designated as red dots; instead, they were assigned as blue dots. Consequently, these misclassified cells are interspersed within the UMAP, diminishing the clarity of the results, and reducing the segregation based on metabolic profiles.

REVIEWERS' COMMENTS

Reviewer #1 (Remarks to the Author):

The authors addressed properly all my queries.

Reviewer #3 (Remarks to the Author):

The authors have revised the manuscript in appropriate ways in response to the reviewers. Specifically, they have removed some text descriptions of findings that are extraneous to the main results, thus shortening the paper as requested by Reviewer 1. The authors have responded to comments of the other reviewers, but without making changes to the manuscript--e.g., they could not increase the number of subjects or recruit an additional study arm. This is reasonable. Some limitations to the manuscript remain, including the small numbers in each group, and the caveats to the PENCIL analysis in terms of predicting protection. However, the manuscript is improved and could warrant publication as now revised. I would just suggest to add a sentence in the Discussion about the limitations of predicting protection, in that exposure to SARS-CoV-2 was not controlled; and that these predictors should therefore be considered preliminary and subject to further validation.

RESPONSE TO REVIEWERS' COMMENTS

Reviewer #1 (Remarks to the Author):

The authors addressed properly all my queries.

Reviewer #3 (Remarks to the Author):

The authors have revised the manuscript in appropriate ways in response to the reviewers. Specifically, they have removed some text descriptions of findings that are extraneous to the main results, thus shortening the paper as requested by Reviewer 1. The authors have responded to comments of the other reviewers, but without making changes to the manuscript--e.g., they could not increase the number of subjects or recruit an additional study arm. This is reasonable. Some limitations to the manuscript remain, including the small numbers in each group, and the caveats to the PENCIL analysis in terms of predicting protection. However, the manuscript is improved and could warrant publication as now revised. I would just suggest to add a sentence in the Discussion about the limitations of predicting protection, in that exposure to SARS-CoV-2 was not controlled; and that these predictors should therefore be considered preliminary and subject to further validation.

We thank the reviewer for the comment, and we amended as requested.

At the end of the discussion, we elaborated on study limitations as follow:

“The study has some limitations. First, the number of patients per group is relatively small, and the investigation is cross-sectional. However, considering the parameters that we have investigated, all groups were homogeneous and there were no outliers. Second, given that the exposure to SARS-CoV-2 was not controlled, the results from prediction analysis should therefore be considered preliminary and subject to further validation. However, our findings suggest that only FTY and natalizumab modify significantly (in terms of phenotype and metabolic status) the SARS-CoV-2-specific B and T cell composition after vaccination”.